# Lassa virus persistence with high viral titers following experimental infection in its natural reservoir host, *Mastomys natalensis*

Chris Hoffmann [1], Susanne Krasemann [2], Stephanie Wurr [1], Kristin Hartmann[2], Elisa Adam[1,3], Sabrina Bockholt[1,3], Jonas Müller[1], Stephan Günther[1,3] & Lisa Oestereich [1,3] ✉

Lassa virus (LASV) outbreaks in West Africa pose a significant public health threat. We investigated the infection phenotype and transmission (horizontal and vertical) of LASV strain Ba366 in its natural host, *Mastomys natalensis*. Here we analyze viral RNA levels in body fluids, virus titers in organs and antibody presence in blood. In adults and 2-week-old animals, LASV causes transient infections with subsequent seroconversion. However, mice younger than two weeks exhibit persistent infections lasting up to 16 months despite antibody presence. LASV can be detected in various body fluids, organs, and cell types, primarily in lung, kidney, and gonadal epithelial cells. Despite the systemic virus presence, no pathological alterations in organs are observed. Infected animals efficiently transmit the virus throughout their lives. Our findings underscore the crucial role of persistently infected individuals, particularly infected females and their progeny, in LASV dissemination within the host population.

Lassa virus (LASV) is a member of the *Arenaviridae* family that is endemic in several West African countries, mainly Nigeria, and the Mano River Union area[1]. In humans this highly pathogenic virus can cause Lassa fever, a severe hemorrhagic syndrome, which is associated with high morbidity and results in up to 18,000 annual deaths, posing a serious public health threat in affected regions[2]. Due to the lack of specific or licensed treatments and preventions, the WHO has categorized LASV as a priority target for vaccine research in their R&D Blueprint[3,4]. The main reservoir host of this rodent-borne arenavirus is the natal multimammate mouse, *Mastomys natalensis*[5]. These commensal animals are one of the predominant rodent species throughout Sub-Saharan Africa[6] and are usually found close to rural human dwellings[7,8]. Furthermore, Mastomys are an important reservoir for several other non-pathogenic African arenaviruses, such as Morogoro virus (MORV)[9–11]. Aside from the main reservoir, *M. natalensis*, several other rodent species, including the closely related Guinea multimammate mouse, *M. erythroleucus*, and several other less commensal

rodents, such as *Mus baoulei* and *Hylomyscus pamfi*, have been identified as additional reservoir hosts for LASV[12,13].

Zoonotic transmission from the rodent reservoir plays a central role for LASV and rodent-to-human contact has been described as the driving factor behind most recent LASV outbreaks[14–16]. Humans contract the virus mainly through direct contact with infected rodents or their excretions[17], although human-to-human transmission has also been observed in a nosocomial context and is associated with a high case fatality rate[18–20]. The strong reliance of LASV and other arenaviruses on zoonotic transmission necessitates a deeper understanding of the virus dynamics in the reservoir host in order to accurately predict and prevent outbreaks. While the virus-host interaction in the rodent reservoir is well described for other arenaviruses like the prototypic Lymphocytic choriomeningitis virus (LCMV), which is considered a model pathogen for virus persistence studies[21], the LASV ecology within its rodent host is still not fully understood, despite the discovery of LASV more than 50 years ago. Most of the current

[1]Bernhard Nocht Institute for Tropical Medicine, Hamburg, Germany. [2]Institute of Neuropathology, University Medical Center Hamburg-Eppendorf, Hamburg, Germany. [3]German Center for Infectious Diseases (DZIF), Partner Site Hamburg, Partner Site Hamburg-Lübeck-Borstel-Riems, Hamburg, Germany. ✉e-mail: oestereich@bnitm.de

knowledge has been derived from field studies with LASV and other non-pathogenic arenaviruses, which highlight the importance of virus persistence for long-term virus viability within the host population[22–25]. However, only few studies have investigated the LASV-host interaction in a controlled laboratory setting. Recent studies have described the LASV dynamics in adult *M. natalensis*. Upon inoculation with LASV, adult Mastomys developed systemic but subclinical infections with detectable viral RNA in blood, urine and various tissues. However, the authors could only observe transient infections, characterized by brief periods of detectable viremia, while viral RNA in organs remained detectable for several weeks[26,27]. Persistent LASV infection on the other hand has so far only been reported once for a small number of animals. The first infection experiments in 1975 demonstrated that Mastomys are susceptible to LASV infection, and that infection of neonates can lead to virus persistence for up to 74 days[28]. Apart from this original study by Walker et al. no other reports of LASV persistence in Mastomys within a controlled laboratory setting exist. In previous infection experiments we could already demonstrate the ability of MORV to establish persistent infections in *M. natalensis* which depended on the age of the animals at the time of infection. The contact with MORV within the first two weeks of life led to virus persistence, whereas older animals only developed transient infections[29].

Based on this previous study with the closely related MORV, we now characterized the viral dynamics of LASV in *M. natalensis* and assessed the impact of the host age on the course of LASV infection. Viral RNA burden in blood and other body fluids, antibody presence in blood, as well as virus titers in organs during transient and persistent infections were analyzed over time. Pathology and virus presence in organs during persistent infections were assessed via histology and immunohistochemistry. Furthermore, horizontal, and vertical transmission between infected individuals and their contacts were investigated.

## Results

### Impact of LASV-infection on animal health

To study the potential development of viral persistence in Mastomys, we inoculated animals subcutaneously (s.c.) with 1000 focus forming units (FFU) of LASV strain Ba366 at age 6–7, 11 or 15 days. Body weight, behavior, and general development, i.e., fur growth, opening of eyes, and motor skill development were monitored every 1–2 days and compared to a naïve control group ($n = 96$). Depending on the litter size, 1–4 individuals per litter were sacrificed weekly for the determination of organ weights and the evaluation of gross pathology, such as malformations, visual bleedings, visual necrosis, or enlarged organs. No adverse effect on the growth, development, or weight gain has been observed following the inoculation with LASV strain Ba366 (Fig. 1a, b, c) compared to naïve individuals (Fig. 1d). Although statistically non-significant, we observed fluctuations in body weight of animals inoculated at 15 days of age during the first week following inoculation. However, animals recovered within a few days from the weight drop. Infected and naïve Mastomys gained on average 0.69 g (SD = ± 0.04) per day during the first four weeks of life. Older animals showed increasing differences in body weight based on sex as previously described[29]. No significant differences in heart and kidney weight were observed between LASV-infected animals and the control group (Fig. 1e, f). Furthermore, no abnormalities in size or structure of other organs or gross pathology were observed during necropsy. The histological analysis revealed no differences between the organs of infected individuals and the uninfected control, and no signs of immune infiltration, cell loss, or tissue disruption in the organs of infected animals were detected (Fig. 2, Supplementary Fig. 1).

### Age dependence of LASV-infection in *M. natalensis*

To characterize the course of LASV-infection in *M. natalensis* and to determine the impact of the host age on the infection phenotype, two litters each were inoculated s.c. with 1,000 FFU of LASV strain Ba366 at the age of 6–7 ($n = 18$), 11 ($n = 36$), 15 ($n = 24$), 28–29 ($n = 17$) or 57–59 days ($n = 21$). Animals were followed for up to 34 weeks post-inoculation (wpi) and viral RNA load in blood, antibody status and virus titers in organs were assessed over time. Following the inoculation with LASV, regardless of their age, animals started to develop viremia at around 1 wpi. Anti-NP-specific IgG antibodies were detected in plasma from 1–2 wpi onwards (Supplementary Table 1). In animals inoculated at 6–7 (Fig. 3a) or 11 days of age (Fig. 3b) viral RNA titers in blood increased in the first 3 wpi to around $10^{10}$ copies/mL and then later stabilized at $10^8$–$10^9$ copies/mL. Despite the presence of antibodies, most animals showed persistent viremia characterized by stable virus titers for up to 34 wpi. Except for four out of 18 individuals, all animals inoculated at 6–7 days remained viremic throughout the experiment. In contrast, following the inoculation at 11 days of age an increasing number of animals were PCR-negative in blood from 5 wpi onwards and only 33% (1/3) of animals were still viremic later than 12 wpi (Supplementary Table 1). LASV could be found in all tested organs of animals inoculated at 6–7 (Fig. 4a) and 11 days of age (Fig. 4b). The highest mean virus titers in inoculated one-week-olds ($10^6$ FFU/g) were observed in lungs, kidneys, and thymus, while animals inoculated at 11 days showed the highest mean virus load ($10^7$ FFU/g) in lungs (Supplementary Fig. 2; Supplementary Fig. 3). Animals either inoculated at 6–7 or 11 days that displayed persistent and stable viremia also showed stable virus titers in organs over time, ranging from $10^3$ to $10^7$ FFU/g. In contrast, the receding or absent viremia observed in animals that demonstrated effective virus control, was always accompanied by virus clearance in organs. The drop in mean virus titers in animals that cleared the infection is shown in Fig. 4a and b, particularly in the group inoculated at 11 days due to the higher proportion of individuals that cleared the virus in this group compared to the group inoculated at 6–7 days.

Following the inoculation with LASV at 15 days of age viral RNA concentrations in blood peaked during the first wpi and continuously declined over time (Fig. 3c). Only 10 out of 14 animals inoculated as two-week olds were still viremic later than 4 wpi (Supplementary Table 1). Initially virus was found in all tested organs, in particular liver, spleen, and lung (Fig. 4c; Supplementary Fig. 4). However, virus titers in organs rapidly dropped in the following weeks. At 7 wpi LASV could only be detected in the brain.

The inoculation of four-week-old Mastomys only caused transient viremia (Fig. 3d; Supplementary Table 1) and infectious virus was found in some organs for up to 2 wpi (Fig. 4d; Supplementary Fig. 5). Similarly, the inoculation at 8 weeks of age also led to transient viremia (Fig. 3e; Supplementary Table 1) and infectious virus was only found in liver, spleen, lung heart, and salivary glands (Fig. 4e; Supplementary Fig. 6).

### Natural horizontal transmission

To assess the natural transmission of LASV from infected Mastomys to their contacts, breeding pairs ($\geq 16$ weeks of age) were continuously co-housed with their litters which have previously been inoculated at 6–7, 11, or 15 days of age. Co-housed individuals (parents and breeding partners, $n = 33$) and the newborn offspring were monitored for LASV presence in blood and organs. Three litters were exposed from birth via co-housing ($n = 23$) to infected older siblings.

Neonates that encountered their previously inoculated older siblings (2–3 wpi) also contracted the virus. No viremia has been detected in neonates younger than one week (Supplementary Fig. 7a; Supplementary Table S2). At 1 week post-birth (wpb) viral RNA was found in the blood of 20% of tested individuals. At 3 wpb all tested animals were viremic. Anti-NP-specific IgG antibodies were detected as early as 1 day post-birth. No infectious virus was found in organs during the first few days of life, whereas at 3 wpb all animals tested positive for LASV in organs (Supplementary 7b; Supplementary Fig. 8). The exposure to LASV had no negative impact on growth or development.

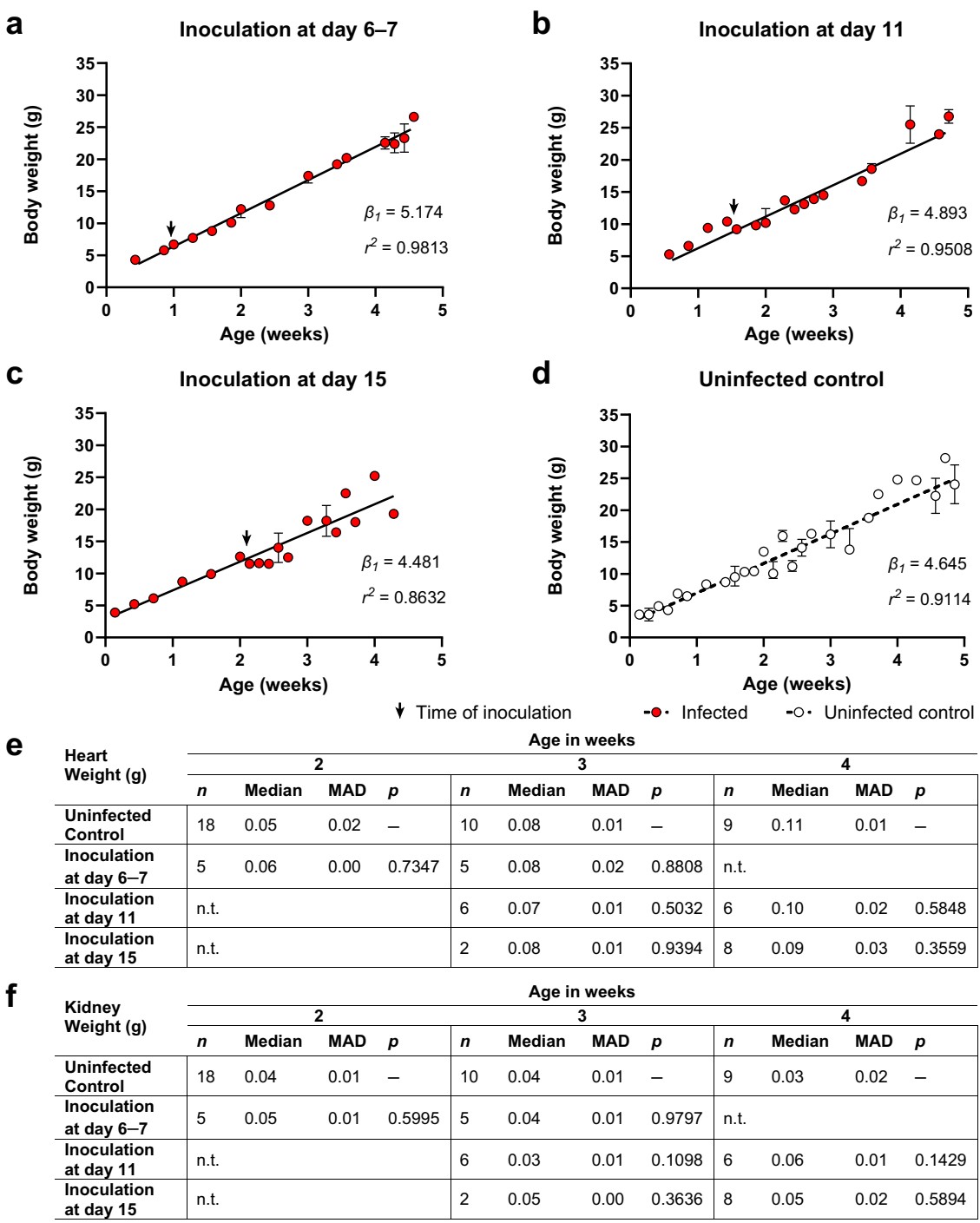

**Fig. 1 | Growth and development of Mastomys infected with LASV strain Ba366.**
Body weight of animals inoculated subcutaneously (s.c.) with 1000 focus forming units (FFU) on day 6–7 (**a**, *n* = 18), 11 (**b**, *n* = 36), or 15 (**c**, *n* = 24) and uninfected animals (**d**, *n* = 96), as well as organ weights of hearts (**e**) and kidneys (**f**) have been measured during the first four weeks of life. The body weight, shown as median with median absolute deviation (MAD), of infected individuals is depicted as red dots, whereas the uninfected control is depicted as white dots. A black arrow marks the time of inoculation. The weight gain in gram per week ($\beta_1$) for each group was determined by simple linear regression. No significant differences in the weight gain of inoculated individuals compared to the naïve control were detected, as determined by one-way ANOVA and *Dunnett's multiple comparison*. Organ weights are depicted as median with MAD. The number of animals euthanized for organ collection (*n*) is shown for each group. Statistical differences in organ weight of infected individuals compared to naïve control animals are indicated by *p* values, as determined by the two-tailed Mann-Whitney test. n.t. = not tested. Source data are provided as a Source Data file.

Exposed individuals did show a body weight increase of 0.87 g per day, which was higher (*p* = 0.0091) compared to the per day growth of the naïve controls (0.66 g per day) (Supplementary Fig. 9a). The parents that were exposed to their inoculated offspring developed antibodies but no viremia or virus presence in organs was detected (Supplementary Table 2).

## Prenatal infection with LASV

To determine whether *in-utero* transmission from an infected female to unborn pups is possible, three pregnant *M. natalensis* females were inoculated approximately two weeks into gestation intra-venously (i.v.) with 10,000 FFU of LASV Ba366. Three litters (*n* = 27 offspring) were born to the inoculated mothers. The inoculation with LASV during pregnancy

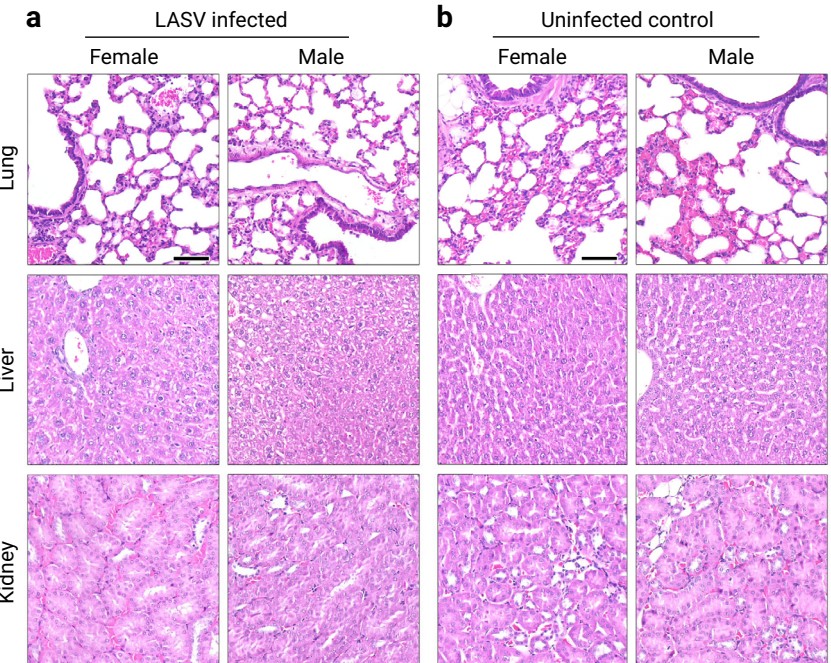

**Fig. 2 | Histological analysis of lung, liver, and kidney.** Organs of female and male Mastomys inoculated s.c. with 1000 FFU of LASV strain Ba366 on day 6–7 (**a**, $n = 5$) were compared to uninfected control animals (**b**, $n = 2$). Infected animals (≥120 days post-inoculation) displayed no signs of immune infiltration, cell loss, or degeneration. Representative images of H&E-stained tissue sections are shown. Black scale bar equals 50 μm.

did not impair the development of the offspring (Supplementary Fig. 9b) or caused tissue abnormalities (Supplementary Fig. 10). Viremia was detected as early as 2 days post-birth, albeit during the first week of life only 29% of animals tested positive for LASV RNA in blood, whereas viremia was detected in all tested animals between 4–8 wpb. Persistent viremia was observed in 60% of tested individuals for up to 40 wpb (Fig. 5a; Supplementary Table 3). Infectious virus was found in organs (Fig. 5b; Supplementary Fig. 11), including stomachs of neonates, as early as 2 days post-birth. The highest titers were observed in the thymus with $10^7$ FFU/g, followed by spleen, heart, testis and stomach of certain individuals with $10^6$ FFU/g. At this early age, not all organs were infected, while at later time points a systemic spread was observed and LASV was detectable in all organs, including prostates, seminal glands, and eyes, for up to 40 wpb. The overall highest mean titers were observed in thymus and brain with $10^7$ FFU/g.

### Vertical transmission from persistently infected females
To assess the occurrence of vertical transmission from females to their offspring; persistently infected females ($n = 4$) were bred with naïve males. These four initial females (parental generation) gave birth to eight litters ($n = 43$). Two persistently infected females of the $1^{st}$ filial generation (F1) were further bred with naïve males. The process was repeated for every subsequent filial generation. By breeding persistently infected females, LASV strain Ba366 was in vivo passaged for five generations in total (F2 generation $n = 81$, F3 generation $n = 8$, F4 generation $n = 12$, F5 generation $n = 5$). Across all generations, pregnant females ($n = 6$) were euthanized within the first two weeks of gestation, and embryos were sampled ($n = 44$).

Infectious virus was already present in placentas, blastocysts and embryos sampled from pregnant females (Fig. 6a, Supplementary Fig. 12). Virus-positive cells were detected in the epithelial layer of the placenta (Fig. 6b). LASV infection in embryos affected all germ layers (Fig. 6c) and infected cells were observed in both epithelial and endothelial compartments. Furthermore, disseminated virus-positive cells were also found in mesodermal compartments.

Offspring born to persistently infected females also became infected with LASV. Only a single individual of 149 animals in total

tested negative for LASV in blood and organs. Viral RNA was detected in blood already 1 day post-birth. Persistent viremia with stable RNA titers ($10^8$–$10^{10}$ copies/mL) was observed for up to 16 months (Fig. 7a; Supplementary Table 3) and LASV-specific antibodies were detectable from 10 days post-birth onwards in the first generation. However, seroprevalence decreased with each subsequent generation, with only 3% of animals from the F2 generation showing a LASV-specific antibody response. No antibodies were detected in blood of individuals from the $3^{rd}$ generation onwards.

LASV could be found throughout all tested organs, including prostates, seminal glands, eyes, as well as cervical and inguinal lymph nodes. The highest mean viral load was observed in kidneys, lungs, and cervical lymph nodes ($10^7$ FFU/g) (Fig. 7b; Supplementary Figs. 13–15). Animals borne to persistently infected females showed little to no fluctuation in virus load over time. The continuous virus presence during early embryonic stages and onwards did not affect general well-being, growth, or development of the offspring and no histological abnormalities were detected (Supplementary Fig. 16). Individuals borne to persistently infected females displayed an increased body weight gain of 0.8 g per day ($p = 0.0066$) compared to naïve controls at 0.66 g per day (Supplementary Fig. 9c).

### Vertical transmission from persistently infected males
To assess the impact of vertical transmission from males through the females to their offspring; persistently infected males ($n = 5$) were bred with naïve females. Pregnant females were placed into clean cages approximately two weeks into gestation, thus the infected males had no direct contact with their offspring. Successful transmission to the offspring fathered by persistently infected males ($n = 92$) was observed in individual animals of five out of 11 litters borne during the experiments. Two of the five infected litters displayed detectable viremia, although not all individuals from these litters tested positive for viral RNA in blood (Supplementary Fig. 17a). Viremia was accompanied by a ubiquitous virus presence in organs (Supplementary Fig. 18; Supplementary Table 3). In the other three infected litters, LASV was only found in some organs and no viremia was detected. Overall, only 17% of all animals fathered by persistently infected males tested positive for

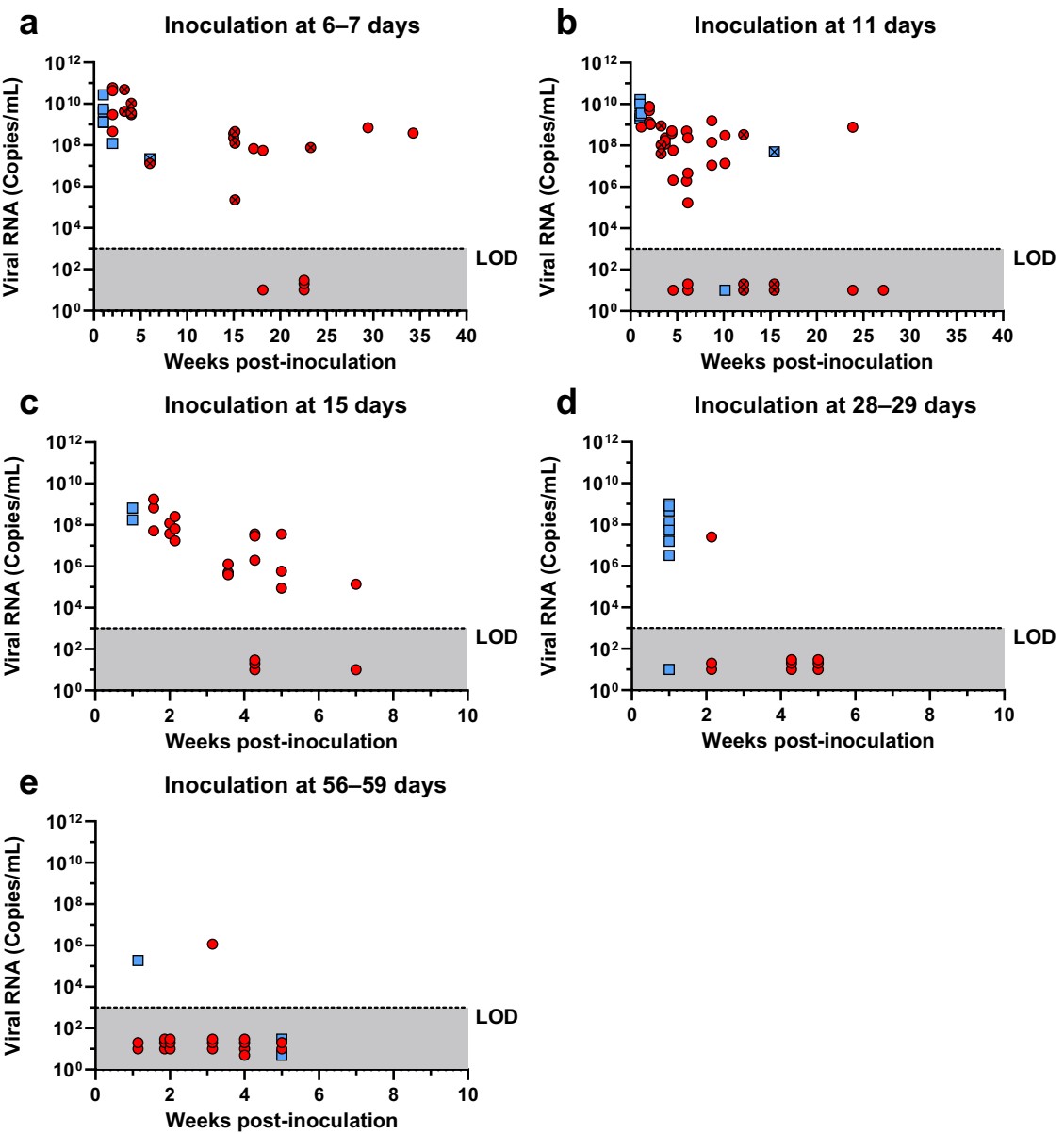

**Fig. 3 | Longitudinal analysis of blood samples from LASV-infected Mastomys.** Virus RNA titers in blood (copies/mL) and serological status are shown over time for animals inoculated s.c. with 1000 FFU of LASV strain Ba366 at 6–7 (**a**, $n = 18$), 11 (**b**, $n = 36$), 15 (**c**, $n = 24$), 28–29 (**d**, $n = 17$), or 57–59 (**e**, $n = 21$) days of age. Blood was sampled at regular intervals and tested for the presence of LASV RNA with qRT-PCR. Ct values were converted into copy numbers using a standard curve. Plasma was inactivated and analyzed for the presence of LASV-specific anti-NP IgG antibodies with ELISA. Blood samples acquired through terminal sampling are indicated by clear symbols, whereas longitudinal samples are indicated by crossed out icons. IgG-positive blood samples are indicated by red dots and IgG-negative samples are depicted as blue squares. The limit of detection (LOD) for the qRT-PCR assay is indicated by the dotted line and dark gray coloration. PCR-negative samples have been assigned a default value below the limit of detection. Source data are provided as a Source Data file.

LASV in organs. Due to the low proportion of infected offspring, the overall mean virus titers in organs were low during the first two weeks of life (Supplementary Fig. 17b). At 4 wpb only two individuals tested positive for LASV in liver or spleen respectively. All naïve females bred with persistently infected males, as well as the resulting offspring tested positive for anti-NP-specific IgG antibodies. The offspring borne during these experiments did not show any differences in body weight gain or development compared to naïve controls (Supplementary Fig. 9d).

**LASV in organs of persistently infected individuals**
Organs from persistently infected animals that were either inoculated with LASV at 6–7 days of age ($n = 5$), borne to females inoculated with LASV during gestation ($n = 4$), or borne to persistently infected females

($n = 27$) were used for immunohistochemical analysis. All tested animals had stable virus titers in blood for ≥120 days. Virus positive cells were detected in almost all organs and tissues. LASV was located amongst others in hepatocytes and epithelial cells of lungs, salivary glands, and kidney tubuli (Fig. 8a). Furthermore, LASV also infected vascular endothelial cells and was also found in the plexiform and inner granular layer, as well as ganglion cells of the retina (Supplementary Fig. 19a). Neurons, endothelial cells in the brain, oligodendrocytes in the cerebellum, and epithelial cells of the choroid plexus also tested positive for the presence of LASV antigen (Supplementary Fig. 19a–d). A strong LASV presence was detected in various tissues of the reproductive tract, such as epithelial cells of follicles and corpus luteum in the ovary, Sertoli cells in testes, epithelial cells in epididymis and prostate (Fig. 8b) as well as the spermatic duct and uterine tube

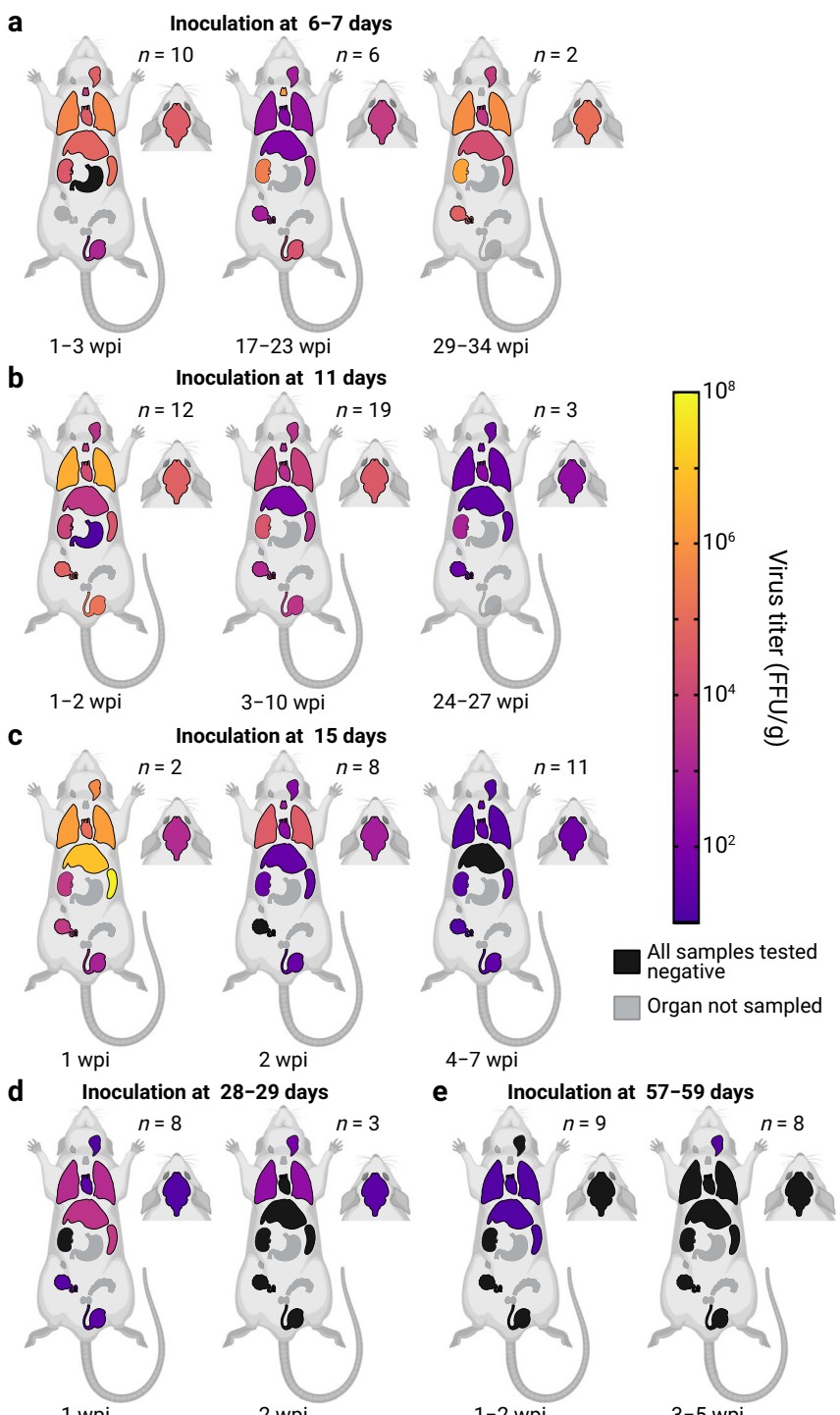

**Fig. 4 | Analysis of organ samples from LASV-infected Mastomys.** Animals were inoculated s.c. with 1000 FFU of LASV Ba366 at 6–7 (**a**), 11 (**b**), 15 (**c**), 28–29 (**d**), or 57–59 (**e**) days of age. The number of animals euthanized for organ collection (*n*) is shown for each sampling period (wpi, weeks post-inoculation). Organ samples were tested for infectious virus via immunofocus assay. Organs are shown as schematic (from cranial to caudal): eyes, brain, salivary glands, cervical lymph nodes, thymus, lung, heart, liver, spleen, kidneys, stomach, inguinal lymph nodes, ovaries, seminal glands, prostate, and testes. Geometric means of LASV-titers in organs are depicted as spectral heat-map. Dark purple: Limit of detection, negative samples have been assigned a default value at the limit of detection. $10^1$ FFU/g; yellow: $10^8$ FFU/g; Black: All samples in the time period tested are negative; Gray: Organ not sampled. Source data are provided as a Source Data file. Created in BioRender. Oestereich, L. (2024) BioRender.com/i19v220.

(Supplementary Fig. 19e, f). No positive signal could be detected in the spleen (Supplementary Fig. 19a). Interestingly, infected cells are not evenly distributed throughout infected organs. We could show that regardless of the target organ, mainly epithelial cells are virus-protein positive. Within one organ, individual positive cells were detected, however, the majority of the persistently infected cells appeared in clusters (e.g. see liver Fig. 8a). Virus protein is abundant in the cytoplasm but may also appear slightly granular. In virus-positive epithelial cells, the majority of signal is detected towards the plasma membrane (e.g. liver Fig. 8a) and often in a highly polarized pattern (e.g. lung or

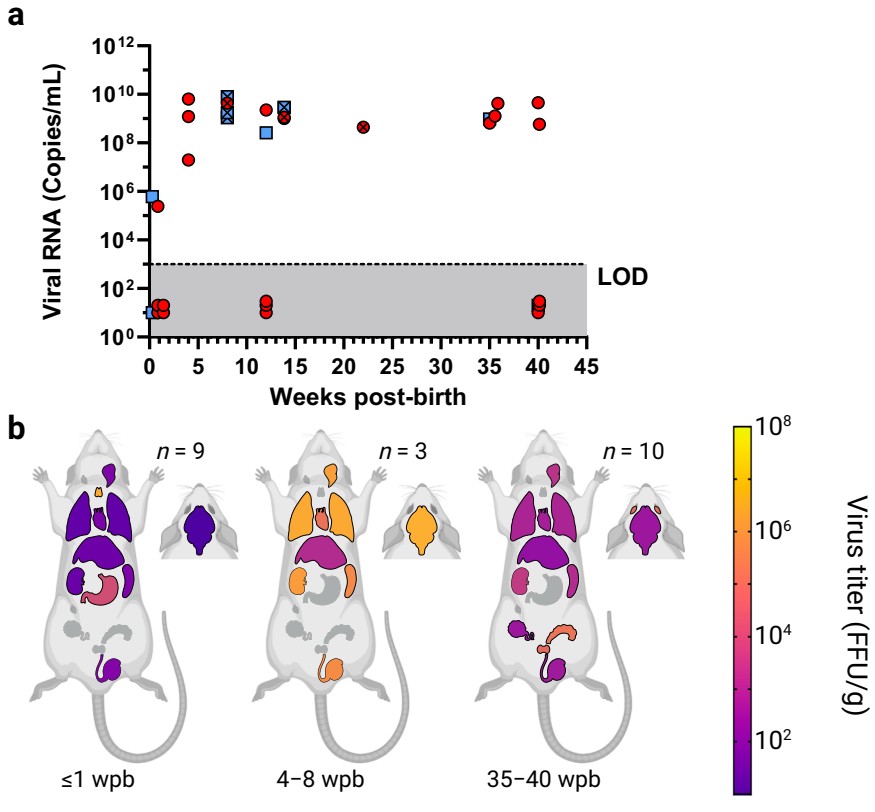

**Fig. 5 | LASV-dynamics in Mastomys following prenatal infection.** Pregnant females were inoculated i.v. with 10,000 FFU of LASV Ba366 roughly two weeks into gestation. The offspring ($n = 27$) borne to the inoculated females were tested for the presence of LASV. Virus RNA titers in blood (copies/mL) and serological status are shown over time (**a**). Blood was sampled at regular intervals and tested for the presence of LASV RNA with qRT-PCR. Ct values were converted into copy numbers using a standard curve. Plasma was inactivated and analyzed for the presence of LASV-specific ant-NP IgG antibodies with ELISA. Blood samples acquired through terminal sampling are indicated by clear symbols, whereas longitudinal samples are indicated by crossed out icons. IgG-positive blood samples are indicated by red dots and IgG-negative samples are depicted as blue squares. Virus titers in organs (**b**) were determined by immunofocus assay. The number of animals euthanized for organ collection ($n$) is shown for each sampling period (weeks post-birth). Organs are shown as schematic (from cranial to caudal): eyes, brain, salivary glands, cervical lymph nodes, thymus, lung, heart, liver, spleen, kidneys, stomach, inguinal lymph nodes, ovaries, seminal glands, prostate, and testes. Geometric means of LASV-titers in organs are depicted as spectral heat-map. Dark purple: Limit of detection, negative samples have been assigned a default value at the limit of detection. $10^1$ FFU/g; yellow: $10^8$ FFU/g; Black: All samples in the time period tested negative; Gray: Organ not sampled. The limit of detection (LOD) for the qRT-PCR assay is indicated by the dotted line and dark gray coloration. PCR-negative samples have been assigned a default value below the limit of detection. Source data are provided as a Source Data file. Created in BioRender. Oestereich, L. (2024) BioRender.com/i19v220.

epididymis Fig. 8a and b). Antibody specificity has been validated through several control stainings (Supplementary Fig. 20; Supplementary Fig. 21).

### LASV in body fluids and excretions
In addition to blood, several other body fluids and excretions have been sampled from animals inoculated at 6–7 days of age (Supplementary Fig. 22a) and from offspring borne to persistently infected females (Supplementary Fig. 22b). Viral RNA has been detected in all collected urine samples with titers of up to $10^{11}$ copies/mL. Furthermore, feces, epididymal plasma, amniotic fluid, as well as oral swabs tested positive for LASV.

The infectivity of sampled body fluids and excretions was assessed in vitro. Urine ($n = 21$) either collected regularly after voiding ($n = 9$) or via bladder puncture upon euthanasia ($n = 16$), oral swabs ($n = 17$), feces ($n = 2$), as well as epididymal plasma ($n = 9$) and amniotic fluid ($n = 2$) were tested for infectivity (Supplementary Fig. 22c, d). Infectious virus was successfully isolated from five urine samples. Four of these samples were obtained by bladder puncture and the remaining sample was collected after voiding. In vitro isolation attempts from samples of terminal bladder punctures showed a higher success rate

(31%) than regular urine samples (11%). Moreover, infectious virus has also been successfully isolated from epididymal plasma (22%) and amniotic fluid (50%). No infectious virus was isolated from oral swabs or feces samples.

### Neutralization capacity of LASV-specific antibodies
Anti-LASV NP-specific IgG-positive plasma samples were tested for the presence of neutralizing antibodies (nAb). Only 11 out of the 88 animals tested possessed nAbs (Supplementary Table 4). Most of these individuals have been inoculated with LASV ($n = 7$), while the remaining animals are the offspring of LASV-infected females ($n = 4$). Despite the presence of nAbs, 72.7% (8/11) of tested animals had still detectable viremia (Supplementary Table 5, 6 and 7). Except for a single individual, nAbs were only detected in samples taken more than 100 days after the initial exposure. The prevalence of nAbs was 62.5% (5/8) among inoculated animals and 12.5% (1/8) for the offspring of infected females between 100–200 days post initial exposure. At more than 200 days post initial exposure, all tested inoculated animals had nAbs, whereas the prevalence of nAbs among the offspring group remained unchanged (2/17). Plasma dilutions at which 50% virus neutralization was achieved ($IC_{50}$) ranged from 1:3 to 1:255.

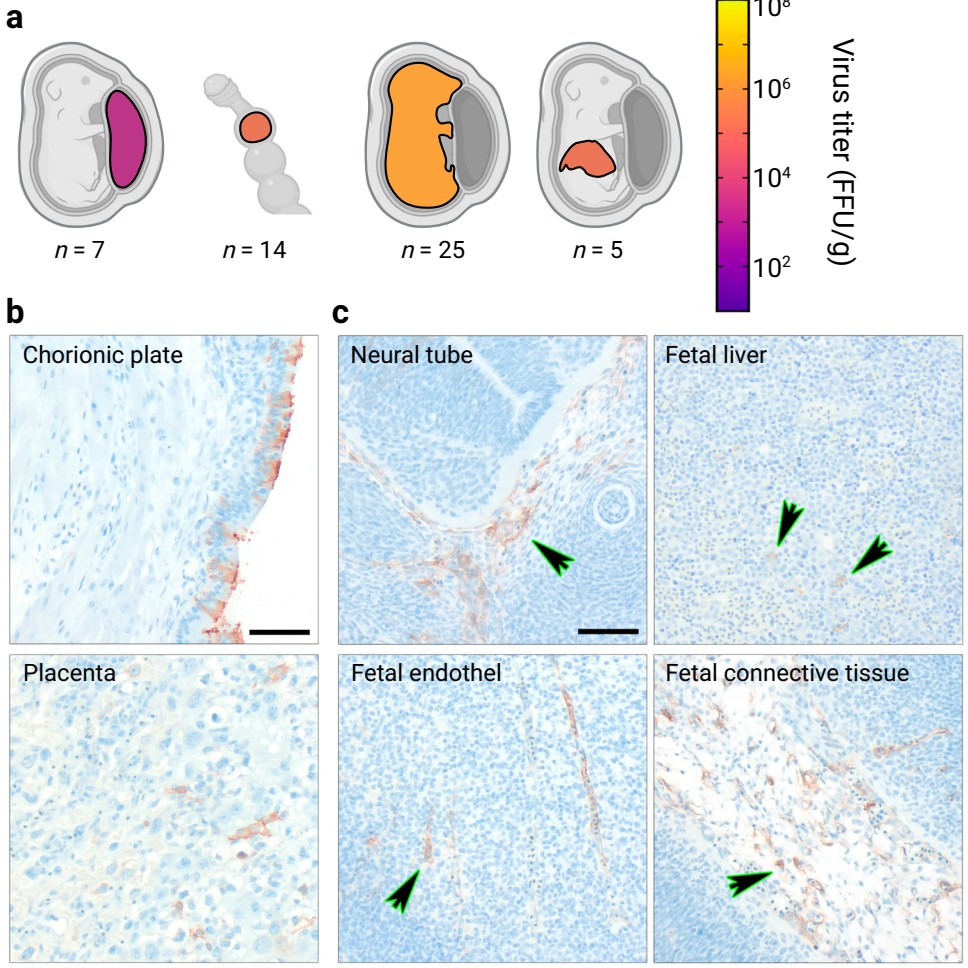

**Fig. 6 | LASV presence in placenta and embryos collected from persistently infected females.** Virus titers in placenta, blastocysts, whole embryos, and fetal livers (**a**) were determined by immunofocus assay. The number of samples (*n*) is shown for each sample type. Immunohistochemical analysis of placentas (**b**) and embryos (**c**) showed infected cells in the epithelial placenta, as well as in embryonic epithelial and endothelial compartments. Furthermore, disseminated virus glycoprotein positive cells were detected in the embryonic mesodermal compartments.

All embryonic germ layers were affected. Geometric means of LASV-titers in organ samples are depicted as spectral heat-map. Dark purple: Limit of detection, negative samples have been assigned a default value at the limit of detection. $10^1$ FFU/g; yellow: $10^8$ FFU/g; Gray: Organ not sampled. Arrows serve as visual aid indicating infected cells. Representative images of the immunohistochemical analysis are shown. Black scale bar equals 50 µm. Source data are provided as a Source Data file. Created in BioRender. Oestereich, L. (2024) BioRender.com/i19v220.

## Discussion

Arenavirus persistence in the rodent reservoir has been described for several arenaviruses[29–34] and evidence for LASV persistence has been found in field studies[22–24]. In concordance with these previous studies, we demonstrate here the ability of the LASV strain Ba366 to establish long-lasting persistent infections in its reservoir host, *M. natalensis*. Since the original study by Walker et al. [28]. we are the first group to have reported persistent LASV infection in *M. natalensis*. Similarly, to what we have observed for the closely related MORV[29], the susceptibility to LASV infection and the development of virus persistence are age-dependent. The inoculation with LASV during the first two weeks of life led to long-lasting infections, characterized by continuous virus presence and shedding in body fluids. With increasing host age at the time of inoculation, the overall duration of infection and the chance for virus persistence gradually declined (Fig. 9). A stark change in the course of infection was observed at the age of two weeks. Inoculated two-week-old animals no longer developed stable persistent infections, but instead cleared the virus, albeit protracted over the course of several weeks. Animals aged four weeks and older only developed very short-lived transient infections, matching observations made in other studies[26,27].

The phenomenon of age-dependent shifts in the infection phenotype is well described for other viral infections such as Hepatitis B virus (HBV) or LCMV and has been linked to changes in the maturing immune system of the host. CD8 + T cell activity especially, has been identified as a crucial factor for the development of immune tolerance mechanisms towards these pathogens and subsequent virus persistence[35–37]. In mice, these shifts in immune activity have been reported to occur around the age of two weeks, which coincides with a changing course of infection for LCMV from persistent to transient infection. In contrast to LCMV infection in neonates, which is characterized by central deletion of virus-specific CD8 + T cells, two-week-old mice can elicit a fully functional CD8 + T cell response. However, unlike in adult mice, this CD8 + T cell response is defined by a lack of expansion and memory T cell generation, leading to a protracted course of infection followed by virus clearance[31]. The age of two weeks is also an important physiological transition phase for Mastomys that is characterized by several developmental milestones, e.g., opening of eyes, leaving the nest, and increasing consumption of solid food. It stands to reason that similar to mice the maturing immune system of *M. natalensis* also undergoes significant changes during this time. Changes in the immune response, notably in the T cell activity, during this maturation process and the development of a central or peripheral

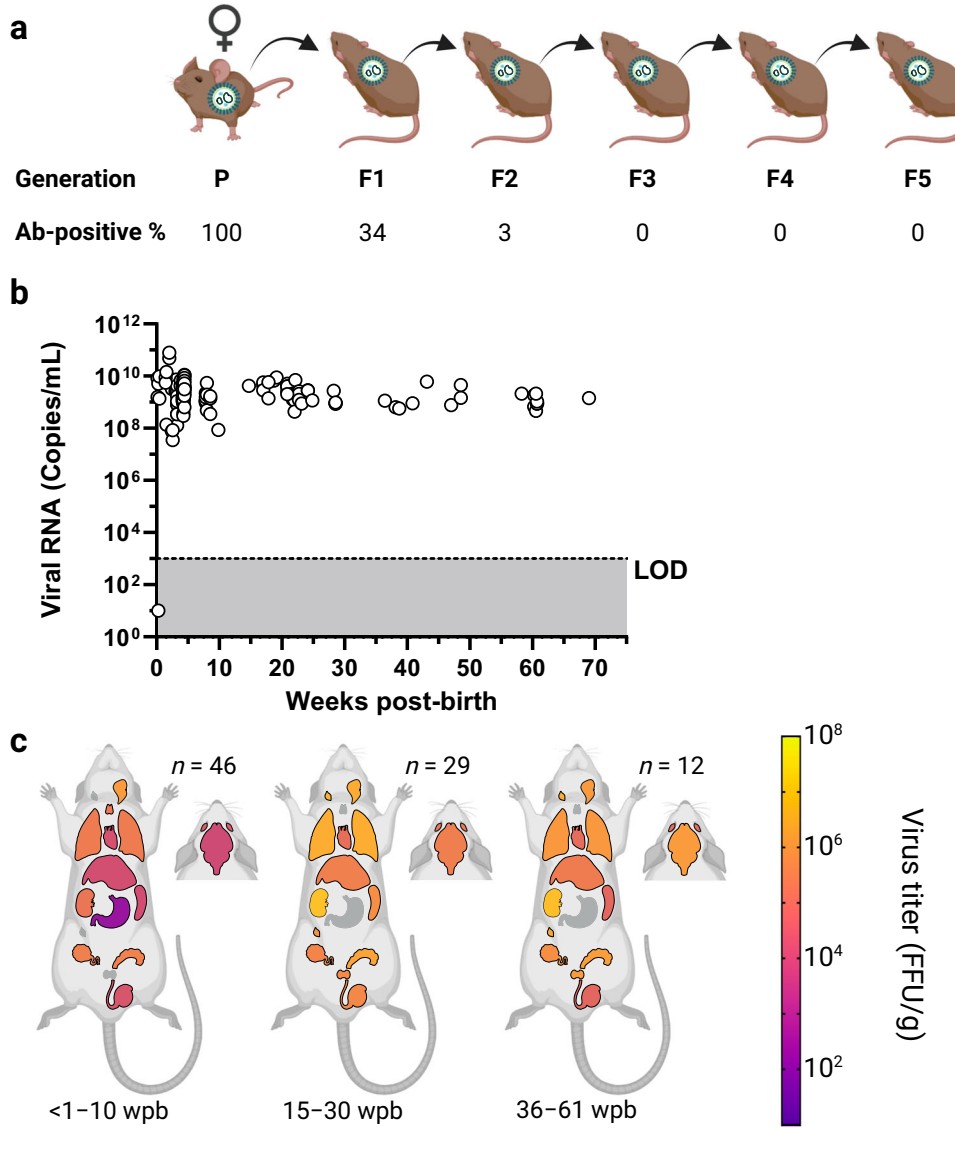

**Fig. 7 | LASV-dynamics in offspring of persistently infected females.** Persistently infected females (parental generation (P)) were bred with naïve males. Females from the resulting litters were further bred. LASV was passaged across five filial generations (F1 to F5) using persistently infected females. **a** Schematic overview of the different generations and the percentage of IgG-positive blood samples. A total of *n* = 149 animals were borne during the experiments. Organ samples were collected from 87 animals. Data for samples from different filial generations (F1–F5) was pooled per time point post birth for the analysis of blood and organ titers. Virus RNA titers in blood (copies/mL) are shown over time (**b**). Blood was sampled at regular intervals and tested for the presence of LASV RNA with qRT-PCR. Ct values were converted into copy numbers using a standard curve. Plasma was inactivated and analyzed for the presence of LASV-specific anti-NP IgG antibodies with ELISA. Virus titers in organs (**c**) were determined by immunofocus assay. The number of animals euthanized for organ collection (*n*) is shown for each sampling period (weeks post-birth). Organs are shown as schematic (from cranial to caudal): eyes, brain, salivary glands, cervical lymph nodes, thymus, lung, heart, liver, spleen, kidneys, stomach, inguinal lymph nodes, ovaries, seminal glands, prostate, and testes. Geometric means of LASV-titers in organs are depicted as spectral heat-map. Dark purple: Limit of detection, negative samples have been assigned a default value at the limit of detection. $10^1$ FFU/g; yellow: $10^8$ FFU/g; Black: All samples in the time period tested negative; Gray: Organ not sampled. The limit of detection (LOD) for the qRT-PCR assay is indicated by the dotted line and dark gray coloration. PCR-negative samples have been assigned a default value below the limit of detection. Source data are provided as a Source Data file. Created in BioRender. Oestereich, L. (2024) BioRender.com/i19v220.

immune tolerance due to the exposure to LASV at a young age[38], could explain the shift in the infection phenotype from persistence towards effective control. Studies in humans and in animal models have already described an association between a robust T cell response and successful LASV control[39–42]. However, further characterization of the innate and adaptive immune response of *M. natalensis* is required to fully explore the underlying mechanisms involved in the development of LASV persistence in its natural host.

LASV infection in Mastomys was characterized by a ubiquitous virus presence throughout all tested body fluids and organs. We frequently detected high virus titers in lungs, kidneys, and gonads of infected individuals suggesting these organs as possible hotspots for virus replication. Furthermore, the observed strong virus presence in the thymus matches in vivo studies with LCMV which have identified thymic involvement as a crucial prerequisite for virus persistence[31,43]. Although we detected virus-positive cells in almost all organs and tissues, epithelial cells were the

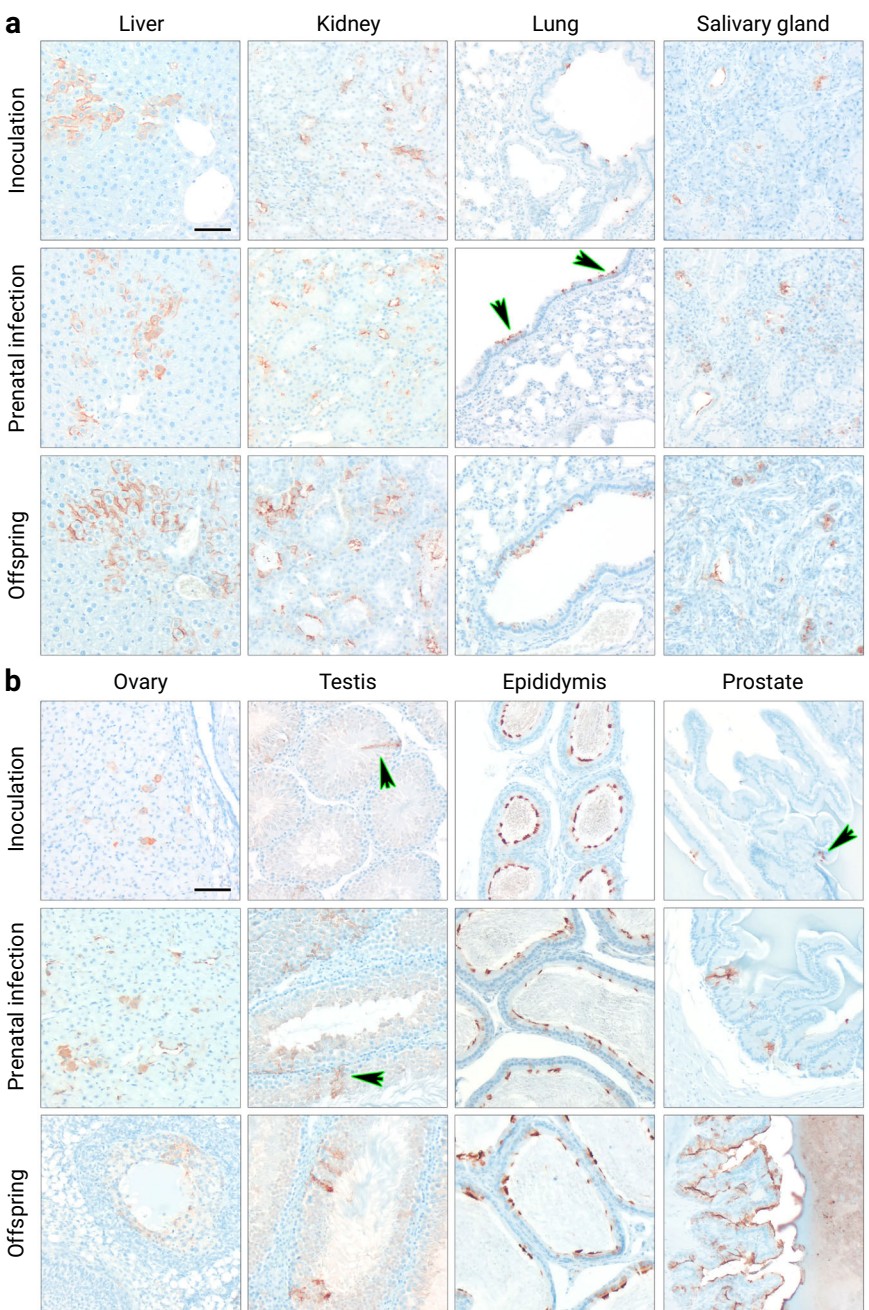

**Fig. 8 | LASV presence in organs of persistently infected Mastomys.** Animals were either inoculated s.c. with 1000 FFU of LASV Ba366 at 6–7 days of age (inoculation, $n = 5$), borne to females inoculated during gestation (prenatal infection, $n = 4$), or borne to persistently infected females (offspring, $n = 27$). Organs were sampled from persistently infected individuals (stable virus titers in blood for ≥ 120 days). Immunohistochemical analysis of liver, kidney, lung, and salivary gland (**a**) shows LASV presence mainly in hepatocytes and epithelial cells. LASV-infected cells were also detected in the reproductive tract (**b**), e.g., epithelial cells of follicles and corpus luteum in ovaries, Sertoli cells in the testes, as well as epithelial cells in epididymis and prostate. Arrows serve as visual aid indicating infected cells. Representative images of the immunohistochemical analysis are shown. Black scale bar equals 50 μm.

main cell type infected during persistence. LASV targeting epi- and endothelial cells has also been described in other immunohistochemical studies with non-human primates (NHP)[40,44]. Of note, while epithelial cells often only showed a faint LASV glycoprotein positive staining in the cytoplasm, a very high amount of virus protein could be detected towards the apical side of these cells. This is especially visible in the epithelial cells of the salivary glands, bronchial epithelium, epididymis, and uterine tube (see Fig. 8a, b and Supplementary Fig. 7f). Thus, we hypothesize that virus particles are produced and secreted by these cells in a polarized manner into the lumen of efferent ducts in these organs. This is in accordance with the high viral loads in body fluids such as urine

and could also explain the degree of vertical virus transmission to off-spring. The observed epithelial tropism indicates the intestines, which have not been sampled during the experiments, as another potential target site for LASV infection and also suggests feces as a possible excretion route. Persistent infections were characterized by stable virus titers in all body fluids and organs over time, whereas effective virus control led to waning virus titers in blood followed by a receding virus presence in organs. This matches the course of infection described by Walker et al.[28]. but stands in stark contrast to our previous observations with MORV infection, where persistent virus replication was limited to certain organs[29].

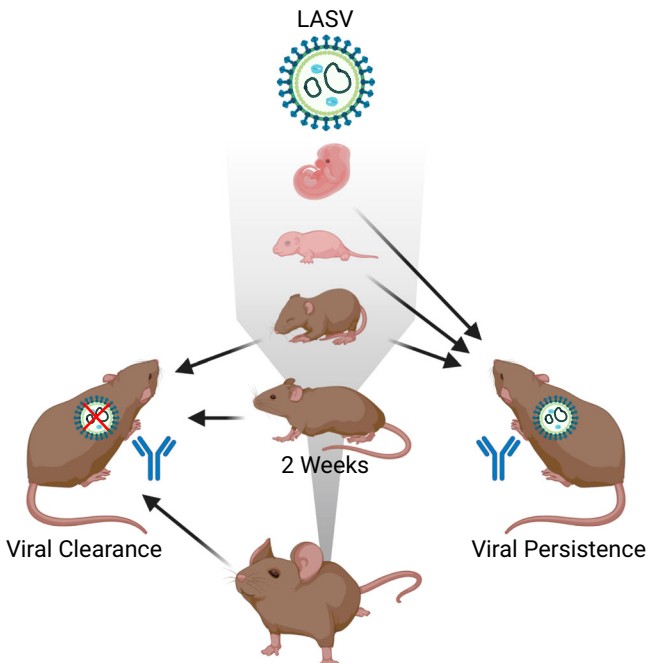

**Fig. 9 | Schematic overview of LASV infection in *M. natalensis*.** The susceptibility of Mastomys to LASV infection and the development of virus persistence are age dependent. Neonates are highly susceptible and can develop life-long infections. Susceptibility to infection and likelihood of virus persistence gradually decreased with age and then drastically declined after the age of two weeks (indicated by the gray gradient). Juveniles younger than two weeks show a mixed phenotype with some individuals exhibiting virus persistence and others effective but protracted virus clearance. Adults and weaned individuals are still permissive to infection but only undergo transient infections. LASV-infected individuals transmit the virus both horizontally to exposed contacts or vertically to their unborn offspring. Infection with LASV led to the production of IgG-antibodies, regardless of the outcome of the infection. Created in BioRender. Oestereich, L. (2024) BioRender.com/i19v220.

The perpetual viremia suggests that transmission through blood is feasible, although, the scarce occurrence of infighting and related injuries in the laboratory setting make it a less likely source for horizontal transmission. Based on the consistently high virus titers in urine and the strong virus presence in kidneys observed during our experiments, shedding through urine seems to be the most likely scenario for horizontal transmission. Furthermore, we were able to isolate LASV from urine samples, demonstrating that in principle virus shed through urine is infectious. Although LASV could be found in salivary glands, no infectious virus was obtained from oral swabs, raising the question whether saliva can serve as transmission medium. Similarly, we were unable to isolate LASV from feces samples, despite the presence of RNA. The encountered difficulties to isolate LASV from body fluid samples match descriptions from other studies using both rodent and human samples[26,45] and further virus isolation experiments are needed to elucidate the role of saliva and feces for virus transmission.

Following the infection with LASV, the first virus-specific IgG antibodies appeared roughly after 1–2 weeks. Similar to our observations with MORV-infected Mastomys[29], there was no correlation between the presence of antibodies and virus clearance. In animals borne to seroconverted mothers, LASV-specific IgG antibodies were detected as early as 1 day post-birth, suggesting the transfer of maternal antibodies. However, the presence of these antibodies did not prevent infection with LASV in the offspring. In concordance with reports on human Lassa fever cases and experiments with NHP[40,41,46], nAbs only appeared very late (≥100 days post infection) in a few individuals and often at low titers. Most of the animals that possessed

nAbs were still viremic and no apparent effect on virus clearance was detected, indicating that nAbs neither conferred protection nor did they impact the course of LASV infection. It is possible that nAbs appear too late and at too low quantities to cause virus clearance on their own without the aid of other immune cells.

The infection of *M. natalensis* with LASV had no detrimental effect on the overall health and development of infected individuals. Although, we did observe a short-lived drop in body weight following the inoculation of animals at 15 days of age which was most likely caused by handling-induced stress during the inoculation procedure. The macroscopic and microscopic evaluations of major organs and tissues of persistently infected individuals revealed no differences when compared to the tissue of uninfected control animals. Detailed analysis of H&E-stained tissue samples showed that none of the investigated organs displayed signs of immune infiltration, cell loss, or degeneration, despite a sometimes very high viral load. This lack of apparent pathology reflects descriptions from previous studies with LASV and the closely related MORV[26–29], and is also in concordance with observations from LASV-infected *M. natalensis* in the wild[47]. Effective virus control in transiently infected animals has previously been associated with short-lived alterations of clinical analytes and histological abnormalities in several organs, such as immune cell infiltrations, interstitial thickening, or tissue disruption in otherwise subclinical animals[26]. It remains to be determined whether these temporary alterations also occur in animals that develop persistent LASV infection.

LASV-infected animals that were co-housed with naïve individuals readily spread the virus to their exposed contacts and even brief transient infections seem to be sufficient for transmission. The course of infections originating from horizontal transmission was dependent on the age at the time of initial exposure, i.e., exposure since birth could lead to virus persistence, while animals exposed as adults only developed transient infections.

The inoculation and exposure experiments indicated animals younger than two weeks as most permissive to infection with LASV and the development of persistence. At this age animals are still nest-bound and heavily reliant on their mothers, making infected females a likely source for infections acquired during early life. The inoculation of pregnant females and the presence of LASV in some of the neonates shortly after birth indicates that transmission to the progeny can in principle occur *in utero*. Studies on LCMV have shown that the timing of infection during gestation is crucial for successful *in utero* transmission and that virus replication in the placenta precedes infection of embryos by roughly 2–3 days[48]. This suggests that the transient LASV infection of the pregnant female is sufficient for the virus to cross the placental barrier but does not necessarily reach all embryos equally before virus clearance in the pregnant female is achieved.

We further assessed mother-to-offspring transmission by breeding persistently infected females with naïve males. During these experiments, a strong virus presence was detected in the female reproductive tract, placentas, amniotic fluid, and embryos, which strongly implies *in utero* transmission as the most likely transmission route. We did not observe any apparent negative effects on fecundity and fertility during our experiment, in contrast to what has been reported for other arenaviruses like Machupo virus (MACV) which can cause embryonic death and resorption in infected *Calomys callosus*[33]. LASV has also been detected in stomachs of suckling pups implying the possibility for transmission via breast milk post-birth or via swallowing of amniotic fluid *in utero*. By breeding persistently infected females and their female progeny we were able to consistently passage LASV across five generations of Mastomys. Throughout these breeding experiments more than 99% of the animals borne to persistently infected females contracted LASV and uniformly developed persistent infections themselves. Our experiments indicate that infected females have an enormous transmission potential. Considering average life

expectancy and overall fertility rates in nature, a single persistently infected female will on average birth 3–4 litters with roughly 8–11 pups during her lifetime[23,49–51], almost all of which will be a life-long perpetual source of infection within the rodent population. The offspring of persistently infected females initially tested positive for LASV-specific IgG antibodies, although, antibodies faded over time in most cases, hinting towards the transfer of maternal antibodies. Once passaged across several generations the humoral response became completely absent. The lack of a humoral response and the absence of effective virus control in these animals further suggests that the exposure to LASV from early embryonic development leads to the acquisition of a central immune tolerance, i.e. the depletion of virus-reactive T and B lymphocytes during early development[38,52]. The absence of an effective or detectable humoral response due to immune tolerance has also been reported for HBV and MACV[37,53].

Virus persistence in the male reproductive tract, in particular testes and seminal fluid, has been reported for LASV and several other viruses[45,54–56]. We repeatedly found high levels of LASV in testes, seminal glands, prostates and epididymis of infected males, suggesting excretions of the male reproductive tract as a possible transmission route to naïve females. We further assessed transmission from persistently infected males to their partners and via the female to their unborn progeny in the absence of direct contact. Although we did observe successful transmission, i.e., confirmed LASV presence in blood and/or organs of the offspring, it was at a much lower rate compared to persistently infected females. Furthermore, the incomplete transmission pattern resembled the one observed following the inoculation of females during gestation. These findings suggest that vertical transmission from father-to-offspring in nature can under the right circumstances occur *in utero* through an infected female that has a short-lived viremic phase, but most likely is predominantly achieved post-natal through direct contact.

In this study, we have been able to observe LASV persistence in *M. natalensis* for up to 16 months. Considering the average life expectancy of Mastomys in the wild rarely exceeds 300 days[49], our findings suggest that LASV can establish true life-long infections in its natural rodent reservoir. In concordance with observations made in the field[23,24,57,58], we could show that horizontal transmission readily occurs between infected animals and their contacts, whereas successful vertical transmission relies on the right circumstances but is highly effective in causing persistent infections in exposed animals. Furthermore, our findings have important implications for the development of potential LASV control and intervention methods that target the rodent reservoir population. We could not only identify neonates as the group most permissive to LASV infection, but our results also emphasize persistently infected females and their progeny as major driving force for long-term LASV survival within the host-population.

## Methods

### Ethics statement
The study was carried out in strict compliance with the recommendations of the German Society for Laboratory Animal Science under the supervision of a veterinarian. All protocols were approved by the Committee on the Ethics of Animal Experiments of the City of Hamburg (N 028/2018, N 050/2021, N 051/2021 and O42/2018). All efforts were made to minimize the number of animals and to mitigate suffering during experimental procedures. All staff members involved in animal experiments and handling underwent the necessary education and training according to category B or C of the Federation of European Laboratory Animal Science Associations (FELASA). All animal experiments in this study are reported in accordance with the ARRIVE guidelines.

### Animals and monitoring
*M. natalensis* used in the study were bred and maintained in the animal facility of the Bernhard Nocht Institute of Tropical Medicine (BNITM).

All animals in the colony are outbred descendants of wild-caught arenavirus-free individuals from Mali that were provided by the Rocky Mountain Laboratories, Montana[59]. Animals were housed in small groups of 3–4 sex-matched littermates or as breeding pairs with litters in individually ventilated cages. Food and water were accessible *ad libitum*. Mastomys were acclimated to BSL4 conditions prior to the experiments. Experiments were performed with whole litters, with female and male individuals being distributed equally across time points for sample collection. Based on the litter size, 1–4 individuals per litter per sampling point were euthanized for terminal sampling. All animals were monitored regularly for general well-being, body weight and clinical signs of disease. Furthermore, general development of neonates and juveniles, including fur growth, opening of eyes, and motor skill development, were compared to a naïve control group (*n* = 96) kept under the same laboratory conditions. Humane endpoint criteria included, amongst others, body weight loss of >10% or reduced growth in neonates and juveniles. Complete scoring and humane endpoint criteria are shown in Supplementary Table 8. Additional information on the animals and handling is given in the Supplementary Methods. Animals were sacrificed for terminal sampling, at the end of the experiments or if any of the termination criteria were fulfilled. Euthanasia was performed via isoflurane overdose followed by decapitation. A total of 538 animals were used in this study.

### Virus strain
The LASV strain Ba366 was obtained from another laboratory[8]. The virus was grown on Vero 76 cells (ATCC® CRL-1587™, American Type Culture Collection, Manassas, VA, USA) and has been passaged less than 10 times in total. Viral Stock titers were quantified via immuno-focus assay, as described elsewhere[42].

### Inoculation
Mastomys juveniles aged 6–7, 11 and 15 days were inoculated subcutaneously (s.c.) with 1000 focus forming units (FFU) of LASV strain Ba366 in 50 μL PBS. Animals aged 28–29 and 57–59 days were inoculated s.c. with 1000 FFU of LASV strain Ba366 in 100 μL PBS. Two whole litters per age group were inoculated in independent experiments. Three naïve adult females were inoculated intra-venously (i.v.) via retro-orbital injection with 10,000 FFU of LASV strain Ba366 in 80 μL PBS roughly two weeks into gestation. Infected animals from these initial infection experiments were used for all subsequent breeding and transmission experiments.

### Transmission experiments
To assess the impact of natural transmission, three breeding pairs were continuously co-housed with previously inoculated offspring, putting pregnant females and subsequent litters into constant contact with virus-shedding individuals. Moreover, breeding experiments were performed using persistently infected individuals. Persistent infection was determined as viremia for at least 80 days with stable virus titers. LASV-infected females (*n* = 4) were bred with naïve males. The resulting female offspring of persistently infected females were further bred with naïve males. LASV was passaged across five generations by breeding persistently infected females. Furthermore, LASV-infected males (*n* = 5) were paired with naïve females. Pregnant females were placed in new cages roughly two weeks into gestation, hence infected males had no direct contact to their offspring. Further information on the inoculation and breeding experiments is given in the Supplementary Methods.

### Sampling procedures
Blood, urine, and organs were sampled at frequent intervals for up to 16 months post-infection, starting with weekly sampling and switching to larger intervals over time. Whole blood, plasma, and urine samples were acquired according to previously established procedures[29]. Oral

swabs were collected from the oropharynx using nylon-tipped swabs (IMPROSWAB®, Improve Medical, Guangzhou, China). Animals were briefly anesthetized with isoflurane for the procedure. The swab tips were placed in 0.2 mL PBS, and vigorously agitated for 30 sec. Feces samples were sporadically collected from animals inoculated at day 6 or from the offspring of persistently infected females. Droppings were placed in 0.5 mL PBS and vigorously agitated for 5 min followed by centrifugation at $2500 \times g$ for 5 min. The supernatant was used for further analysis.

Heart, liver, spleen, kidney, lung, brain, and sublingual as well as submandibular salivary glands were collected from all euthanized animals. Stomachs were taken from animals younger than two weeks, while gonads (ovaries and testes) were collected from individuals older than one week. The thymus was sampled from animals inoculated within the first two weeks of life or animals borne to infected females. Eyes, prostates, seminal glands, epididymal plasma (cell-free content of the caudal epididymis), as well as cervical and inguinal lymph nodes were collected from selected animals borne to infected females. Furthermore, amniotic fluid, placentas and embryos were taken from pregnant infected females (less than two weeks into gestation) following euthanasia. Sampling procedures for the collection of amniotic fluid and epididymal plasma are described in more detail in the Supplementary Methods.

### Determination of virus titers and antibody status

The presence of viral RNA in whole blood, urine, oral swabs, feces, amniotic fluid and epididymal plasma was determined by qRT-PCR assays. Samples were inactivated and RNA was extracted using the QIAmp Viral RNA Mini Kit (QIAGEN, Venlo, The Netherlands) according to the manufacturer's instructions. PCR reactions for the detection of the LASV L gene were set up using the RealStar® LASV RT-PCR Kit 2.0 (altona Diagnostics, Hamburg, Germany) according to the manufacturer's instructions. In vitro transcripts based on the LASV strain Ba366 L sequence were used to create a standard curve. The limit of detection (LOD) of the qRT-PCR ($10^3$ copies/mL) was set based on the lowest measured titers of quantified in vitro transcripts. Negative samples were assigned default values below the LOD.

Plasma samples were inactivated by mixing with an equal volume of PBS containing 2% Triton X-100. The presence of Anti-LASV NP-specific IgG was assessed with a capture enzyme-linked immunosorbent assay (ELISA) using the BLACKBOX® LASV IgG ELISA Kit (Diagnostics Development Laboratory, Hamburg, Germany) as described elsewhere[60].

Collected organ samples were homogenized in 1 mL cell culture medium using the FastPrep-24™ 5 G tissue lysis system with the Lysing matrix D. Infectious virus titers in organs were determined by immunofocus assay as described before[42]. The LASV NP-specific monoclonal antibodies L2F1 or 2B5[61] were used to detect infected cell foci. The LOD of the immunofocus assay ($10^1$ FFU/g) was set based on the lowest measured virus titer after taking organ weight into account and negative samples were assigned the default value of the LOD.

### Histological and immunohistochemical analysis

Organs from persistently infected individuals, with stable virus titers in blood for ≥120 days, were used for histological and immunohistochemical analysis. Mastomys tissues were thoroughly fixed in 4.5% formaldehyde solution (SAV Liquid Production GmbH, Flintsbach am Inn, Germany) for 72 h at RT to inactivate the virus, dehydrated, and processed for paraffin embedding. Sections (2 μm) were subjected to H&E staining according to standard procedures. Immunohistochemical detection of LASV-infected cells was performed as follows: Tissue sections (2 μm) were subjected to antibody-specific antigen retrieval using the Ventana Benchmark XT (Ventana Medical Systems, Tuscon, AZ, USA). Sections were blocked in PBS with 10% rabbit serum and afterwards incubated with the human primary LASV GP specific

antibody 22.5D (Ab00225D, Zalgen, Frederick, MD, USA; 1:100 dilution in PBS with 10% rabbit serum). An anti-human secondary antibody was used to detect specific staining and visualized with 3,3'-Diaminobenzidine (DAB) substrate using the ultraView Universal DAB Detection Kit (Ventana Medical Systems, Tuscon, AZ, USA). Tissues were then counterstained with hematoxylin. Images were acquired with the Leica DMD108 digital microscope. Representative images of the IHC and H&E staining shown throughout this study are matched, i.e. taken from the same individuals, within the different experimental groups. Detailed information on the animals for which IHC and H&E images are shown is available in Supplementary Table 9.

### In vitro infectivity assays

For the assessment of the infectivity of body fluids collected during in vivo experiments Vero cells were seeded on 12 mm glass coverslips in 24-well plates and inoculated with 200 μL of 2–3-fold dilutions of body fluid samples that previously tested PCR-positive for LASV. Following inoculation at 37 °C and 5% $CO_2$ for 1 h, the inoculum was removed and replaced with fresh cell culture medium. Cells were incubated at 37 °C and 5% $CO_2$ for 3 days. The cell culture medium was removed, and cells were fixed by placing the coverslips in 4.5% formaldehyde solution for 30 min. The LASV NP-specific monoclonal antibodies L2F1[61] or the polyclonal Rabbit-LASV Nucleoprotein antibody (GTX134884, GeneTex, Inc. Irvine, CA, USA) were used to detect LASV-infected cells. Fluorescein-conjugated AffiniPure Goat Anti-Mouse IgG (Jackson Immuno Research, West Grove, PA, USA) or DyLight™ 488 Donkey anti-rabbit IgG (BioLegend®, San Diego, CA, USA) were used to visualize infected cells. Additional information on the tested samples is given in the Supplementary Methods.

### Neutralization capacity of LASV-specific antibodies

Information on the tested samples can be found in the Supplementary Methods. Vero cells were seeded on white 96-well plates ($1.2 \times 10^4$ cells/well) the day prior to the experiment. Plasma samples of animals infected with LASV were complement-inactivated at 56 °C for 30 min followed by centrifugation at 10,000x g for 10 min. LASV strain Ba366 was diluted in DMEM with 1% FCS (300 FFU/well) and pre-incubated 1:1 with serial dilutions (1:8 to 1:256) of the plasma samples for 1 h at 37 °C and 5% $CO_2$. The virus-plasma solution was then used to inoculate Vero cells for 1 h at 37 °C and 5% $CO_2$. The inoculum was removed and replaced with 1%-methylcellulose medium overlay. Cells were incubated at 37 °C and 5% $CO_2$ for 24 h. Inactivation and subsequent staining were performed according to the immunofocus assay described elsewhere[42]. The iSPOT Elispot reader Version 7 (AID, Strassberg, Germany) was used to count stained foci. The percentage of foci reduction compared to an untreated control (incubation with only virus) was calculated using the mean numbers of counted foci. Nonlinear regression analysis was performed to calculate the $IC_{50}$.

### Statistics and data presentation

Data presentation and plot preparation was done in GraphPad Prism 9. Weight data was tested for normality using the Shapiro-Wilk test and QQ-Plot analysis. Body weight gain per day was determined by simple linear regression. Differences in average body weight gain between experimental groups and the naïve control were determined by one-way ANOVA and *Dunnett's multiple comparison*. Differences between organ weights were determined by the Mann-Whitney test. Nonlinear regression analysis using the sigmoidal 4PL regression model was used to calculate the $IC_{50}$ for neutralization assays. Heatmaps of organ titers were created in GraphPad Prism 9 and transferred to the anatomical schematics with Biorender.com. Histological and immunohistochemical images were arranged with Adobe Photoshop (version 24). All Organ schematics and the overview figure (Figs. 4, 5, 6, 7, 9, Supplementary Fig. 7, and 17) were created in Biorender. Oestereich, L. (2024) BioRender.com/i19v220, accessed on 13 September 2024.

**Reporting summary**

Further information on research design is available in the Nature Portfolio Reporting Summary linked to this article.

## Data availability

The data generated in this study are provided in the Supplementary Information and in the Source Data file. Source data are provided with this paper.

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

## Acknowledgements

This research was funded by the German Research Foundation (DFG; grant numbers: GU 883–1 and GU 883-2) awarded to S.G.. L.O. received funding from the Leibniz Association (grant number: J59/2018). We thank Heinz Feldmann and Kyle Rosenke from the Rocky Mountain Laboratories, RML, Hamilton, USA for providing the breeding stock of *M. natalensis*. We further thank Yvonne Richter and all the other staff in the animal facility at the BNITM for their support with the *M. natalensis* experiments and for taking excellent care of the breeding colony.

## Author contributions

Conceptualization, C.H., L.O.; methodology, C.H., S.K.; formal analysis, C.H., S.K., S.W., K.H., E.A., S.B., J.M.; investigation, C.H., S.K., L.O.; writing of the original draft preparation, C.H.; writing of review and editing, S.K., L.O.; visualization, C.H., S.K.; supervision, L.O.; project administration, S.G., L.O.; funding acquisition, S.G., L.O.. All authors have read and agreed to the published version of the manuscript.

## Funding

## Competing interests

The authors declare no conflict of interest. The funders had no role in the design of the study; in the collection, analysis, or interpretation of data; in the writing of the manuscript, or in the decision to publish the results.
