## [Peer Review file · Nature Communications]

Lassa virus persistence with high viral titers following experimental infection in its natural reservoir host, *Mastomys natalensis*

Corresponding Author: Dr Lisa Oestereich

Version 0:

Reviewer comments:

Reviewer #1

(Remarks to the Author)

Hoffmann, et al report on a much-needed systematic analysis of the persistence of Lassa virus (LASV) in one of the natural reservoirs for the virus; *Mastomys Natalensis*. The authors provide excellent data on the numerous ways that LASV persistence can occur. Of particular note is the ability of *M. Natalensis* to be persistently infected with LASV for over 16 months in a laboratory setting which goes well beyond the normal life span of these rodents. These data along with the horizontal and vertical transmission (male or female) shown in this manuscript brings home the importance of how much and for how long the transmission threat comes from *M. Natalensis*.

Minor comments:

Line 79: What is the inoculation route? Could not find in methods either.

Antibodies surveyed within the study were directed at the nucleoprotein. Are there antibodies to the glycoprotein found and are they neutralizing at all?

Line 185: Why the change in inoculation route for the prenatal infection model?

Reviewer #2

(Remarks to the Author)

Understanding how zoonotic arenaviruses such as LASV are transmitted and maintained in reservoir hosts is important to predict and prevent outbreaks. Only a few studies have looked at persistence of LASV in its natural host in laboratory experiments – especially young animals. This study examines both horizontal and vertical transmission of LASV in its natural host rodent species.

The work presented here is quite a thorough tour-de-force study of persistence in the natural reservoir species. In adults and animals older than 2-weeks, LASV infection resulted in transient infections with subsequent seroconversion. However, the authors of the study here found that neonates and younger juveniles exhibited persistent infections lasting up to 16 months despite antibody presence. Intriguingly, despite detection of viral antigen in many tissues, no pathological lesions were found, and no clinical signs were observed in the animals. The persistently infected animals were able to transmit the virus throughout their lives, providing a new understanding of transmission dynamics in the natural host. The antibodies produced during infection appear to be mostly non-neutralizing, which provides an intriguing avenue for future exploration. The results here are similar to this research team's work with the related virus MORV whereby reservoir contact with MORV within the first two weeks of life led to virus persistence, whereas older animals only developed transient infections.

This is a very well-done study using a difficult experimental system (BSL-4 virus; wild rodent species) that provides important new information for the field. I particularly like the mouse organ heat maps in Fig 4-7.

My comments are minor:

1. How were LASV inoculation doses chosen?
2. Fig 5A – does each symbol represent a single animal tested at that time point, or were some animals retested longitudinally?
3. Fig 5 – can the authors confirm that transmission happened during pregnancy rather than after the pups were born?
4. At what stage of gestation were the pregnant females euthanized (Line 221-222 and Fig 6)?
5. Arrows on histo images in Fig 6C would be beneficial.
6. What counterstain is used in the IHC images?
7. Line 536-537 – virus stock used for these studies has an unknown passage history. Has it been sequenced?
8. Fig 9 on the transmission model is confusing.

Reviewer #3

(Remarks to the Author)

Key-findings

Hoffman et al., Lassa virus persistence with high viral titers following experimental infection in its natural reservoir host, *Mastomys natalensis* describes the results from a great number of experimental infections. The key result from these experiments is that virus infection that starts prior to the age of 2 weeks causes the mice to become persistently infected, while infection that starts after this age will be transient. Another key finding is that some infected animals remain so while they develop antibodies, including neutralizing antibodies. Lastly the virus infection could not be associated with disease in any of the infected animals.

Quality of the data:

The animal experiments seemed to have been performed with great care, and analyses thorough and detailed including serial and long-term clinical assessments, virology (qPCR and culturing), serology (ELISA and VNTs) and histopathology including IHC. I really like LASV was investigated in its original host. This type of research can help to understand the virus infection in the reservoir much better and hence help in preventing disease but also understanding the disease in humans - comparative pathogenesis. It is often not easy to do these experiments in non-conventional animals in the laboratory. It is also not easy to follow up on the long time (16 month) so congratulations with this success. However, I do have some concerns about the set-up of the animal experiments.

- Virus strain: Was any sequencing performed to learn how the virus strain used for these experiments (that has an unknown passage history) was comparable to a wildtype virus – and thus can be used to model infection in the wild?

- Route of inoculation: this can have great effects on type and course of virus infections. Subcutaneous and intravenous routes were used, while authors themselves argue that this is not likely to be an important natural route of infection for LASV in *Mastomys* mice.

- Read-out: H&E and IHC were performed on persistently infected animals – at varying times post inoculation. If any conclusions are drawn by authors regarding a lack of tissue response to virus, authors should be very clear about which groups were actually investigated by these techniques. In my opinion these groups should be more carefully defined because there are several different variables that are likely to influence the pathogenesis or pathology, and therefore they should not be lumped into one 'persistently infected' group. Relevant are at least the age at inoculation, duration of the infection, antibody presence at time of death.

- A lot of data is presented- an impressive amount of work. However, I wonder if all are necessary, or necessarily in the right format to show the main outcomes of the study. I got lost between the different experiments and read-outs and data -see more detailed comments below. What was the aim and outcome of the transmission experiments? It did not become clear to me.

Despite my concerns I think the main conclusion on the age of inoculation determining the likelihood of chronic LASV infections is correct based on the research findings described here. I welcome an explanation for the inoculation routes used – and why the infections in the experiment mimic the natural situation despite of it. The histology and immunohistochemistry photos clearly show the occurrence of widespread infection without clear development of lesions in those cases; however it is not clear what cases these examples represent. This could be made more clear.

In my opinion the main result, the effect of the age when *Mastomys* rats become infected, determining the further development of the infection, is significantly contributing to our understanding of the LASV reservoir.

Also, I have some more specific issues with clarity and data presentation, or questions /concerns, outlined below:

Please explain, if one aim of the experiments was to determine viral excretion sites and routes of transmission, why was feces tested, but not intestines (while the virus is obviously epitheliotropic). Could feces not be an excretion route, via intestinal epithelial cell infection?

Regarding the feces that was tested, there were only two (line 324); while 538 animals were used in the experiments? Not clear how difficulties of testing feces for infectious virus were overcome (as the sample usually negatively affects the cells on which virus is grown).

Line 18 and 19: 'However, neonates and younger juveniles' what is younger than a neonate? Consider to rephrase 'neonates and younger juveniles' to something like 'mice younger than 2 weeks' to be clear.

Line 43: Please rephrase, as zoonotic transmission itself can not be driving an outbreak; people making contacts with animals can be seen as driving the occurrence of zoonotic transmissions, which can be the cause of the start of the outbreak.

Line74: 'organ pathology'

Line 90: size weight would be more precise here

In figure 1 1-d: body weight on y axis instead of 'weight' would be more clear.

In figure 1c I would challenge the statistics that there is no effect seen from the inoculation. Five dots after the challenge are below the black line indicating the trend, before weights recover again. Then there is also more variation in week 3.5-4.5 with dots being well above or below the black line in this group – more than in the other groups. Not certain what statistics should be used but I imagine one could calculate based on the other data how likely it is to have by just variation, those five dots in a row after the inoculation lower than expected based on the growth curves seen in all groups. Still, the effect is not necessarily from the infection, could also be from the inoculation as the negative control group were not sham inoculated.

Fig 2: Please provide a serial section with immunohistochemistry staining to show the presence of virus antigen in cells and tissues provided. As lesions are mostly expected to be associated with infected cells it is important for the interpretation to show them side by side.

In general for clarity, but eg Line 117: 'virus load was determined' it is important to make a difference between viral load measured as RNA by PCR, or as infective viral particles by virus titrations, especially as you performed back calculations to copies/ml instead of reporting ct values it might not be clear what was actually measured and this is relevant for interpreting 'persistent infection' as being individual shedding infectious virus, or individuals being viral RNA positive for a very long time.

Line 123, four individuals of how many in total?; also this sentence seems to contradict with the previous one as apparently not all animals remained viremic upto 34 weeks.

Lines 132 /133: 'Some animals exhibited effective virus control and receding, or absent viremia was followed by virus clearance in organs.' Not sure what the results are that are described here. Some? How many of the total? The previous sentence suggests only persistent infected animals are described from that point onwards. How was decided they were persistently infected with absence of viremia measured?

Also, from the methodology I understood that for tissue samples viral loads were determined by virus titrations and hence testing for infectious titers. Would be important to have results on how infectious titers and ct values compared in these organs of animals with antibodies to those without antibodies – to see if there was neutralization observed when culturing from the tissues of animals in which antibodies were detected.

Line 144: 'LASV found in some organs for upto 2 weeks.' This is an example. Please throughout the results, including in figures, make clear what methodology was used to state virus presence, with IHC virus antigen, with PCR viral RNA or with culturing infectious virus? Or multiple techniques? This matters a lot for interpretation. For example RNA can be often found after viral infections while it is then not corresponding to aversive health effects or transmission.

Fig 3: would prefer to see the a to e figures labeled with the group ID (inoculation time)

Line 145: not clear what authors seem as 'limited' virus presence and what this is based on.

Fig 4: Nice graphical way to show virus distribution in the different groups over time. However, what is the aim of the figure? I would think the main result shown is which cells or tissues contribute to the viral load present in the infected individual? Why was virus titer used as data for the figure, and not immunohistochemistry? IHC can be much more directly linked to virus in tissue. Immunohistochemistry is easier to interpret in relation to tissue, as with the other techniques you are also measuring virus in fluids. It is possible to quantify immunohistochemistry results in many ways. What are the arguments to use virus titer instead? Could the results be influenced by the presence of antibodies in some of the animals?

Visually: the difference between blue and black is not clear to me, why not just use black or blue as the interpretation is the same (no virus detected). For eyes it is not possible to see the color to see the viral titer. Again, I would prefer to have the group indication (so the inoculation time) indicated in the separate figures for clarity. How to deal with male/female and gonads in the figures as the n will be different for those organs.

Line 172: 'LASV-infected individuals' please indicate at what time post inoculation the infected individuals are, and how the exposure was.

Line 179: 0.87 g per day, $p = 0.0091$; please indicate if this was 0.87 g per day growth, or 0.87 g per day extra growth compared to the non-exposed?

Line 180: 'The parents of inoculated offspring showed only signs of a transient infection' what does this mean? Clinical signs? What? Or RNA detection in blood? What is meant with 'inoculated' here, I thought they were infected by exposure to infected animals?

Line 215: it is not clear where these 'persistently infected' females come from. Were these from the group that was inoculated at day 6 post partum?

Lines 291 and further: for interpretation it is important to describe the distribution of the virus positive cells in the tissues (individual, clusters, diffuse etc.). Also important to describe the distribution of the staining in different cell types (granular or diffuse, cytoplasmic etc.).

Line 336: 'Despite the presence of nAbs, 72.7 % of tested animals had still detectable viremia (Table S17, S18 and S19).' For each percentage add number positive, and numbers tested (8/11).

Lines 400-402: 'Of note, the staining of the viral glycoprotein showed a highly polarized virus protein production in epithelial

cells towards the lumen of the respective organs. Not sure to what results this is referring to? Which organs was this observed? Was it polarized within the cell? Or within the tissue? Many of the described epithelial cells are just one cell layer lining the lumen...

Line 431: 'Antibodies with other non-neutralizing effector functions could, however, still contribute to virus clearance.' This confuses me. Was more viral clearance seen in individuals with non-neutralizing antibodies? Or what is this sentence referring to or explaining?

Line 581: 'Determination of virus titers and antibody status' There is lots of comparisons between groups on virus titers per organ e.g.. How was the input controlled to make this comparison valid? E.g. weighing of the sample size or RNA output? There is no data provided on how the organ samples were pretreated (e.g. homogenizing etc) – although information of feces and swabs is present in 'sample pretreatment'.

Lines 467-469: 'LASV has also been detected in stomachs of suckling pups implying the possibility for transmission via breast milk.' I don't see a reference for that? Please be aware that pups in utero will drink amniotic fluid, and that could also be an explanation for finding virus in the stomach.

Starting line 480: 'suggesting that the exposure to LASV from early embryonic development leads to the acquisition of a central immune tolerance that also encompasses LASV-reactive B cells.' Please explain the term 'central immune tolerance' or provide a reference. And later also 'immune tolerance mechanisms'. What mechanisms? It is not clear how 'Once passaged across several generations the humoral response became completely absent' suggests 'the exposure to LASV from early embryonic development leads to the acquisition of a central immune tolerance that also encompasses LASV-reactive B cells.'

When virus replicates in a fetus in utero prior to positive and negative selection of T and B cells, the T and B cell repertoire might not recognize the virus proteins as being non-host, and therefore no antibodies nor T cell immune response will occur. Is that what you are referring to? Please explain more, and also provide an explanation for how this increases with each passage.

Paragraph starting line 484: I don't understand the comparison. Is the question not rather if males can infect adult females via semen, and via adult females that are pregnant, fetuses can become infected? I don't understand how transmission from semen to offspring is relevant if the role of the transmission to the mother is not considered.

Authors do not discuss or explain how some animals with detectable neutralizing antibodies can remain infectious over a long time. Does the virus escape immunity by hiding at immune privileged sites and re-infecting other sites from there? Does the virus escape immunity by mutations?

Supplementary data

Line 31: 'Two whole litters per experiment group were inoculated to account for the genetic variations of the parents.' I don't think that two breeding pairs can be seen as accounting for genetic variations. Rather I would state that animal experiments for ethical reasons should be balanced and that you have designed the experiments with the assumption that these two breeding pairs could be seen as representative.

lines 35-36: 'Persistently infected females used in the breeding experiments originated from either day 6 inoculation or the inoculation of a pregnant female' I have missed the description of infection dynamics after inoculation during pregnancy – how long were these followed for virus presence? Does inoculation during pregnancy consistently result in a chronic >40 week infection? Describing the infection dynamics after inoculating pregnant females is quite an important. These females were >12 weeks old, and it doesn't fit the pattern described in the other parts that they become persistently infected.

Table S2: Why only min-max, and not an average or mean provided? Rather than a table I would like to see these kind of data in bar charts, similarly organized (per organ, and per group – together) so that the viral loads over time for each organ becomes clearer.

Table S5: It is not 'natural', it is an experiment. 'Exposure to infected parent since birth' or something similar is more precise. 'Exposure of adults to infected individuals' Adults at what age(s)? What is not clear for this experiment is: At what stage(s) of their infection were the individuals used to expose the adults? What were their excretion routes, of those selected infected individuals? What was their immune status at the moment of the exposure experiment? How long were adults exposed to the infected individuals and via what routes? How come the data from 16 to 52(!) weeks is now lumped together?

For table S8: please provide information in the legend on what organs were tested.

Table S10: chronic male and chronic female are used, as well as persistently infected parent. Please be precise and consistent with terminology. It is not clear what the table is referring to. How can one differentiate between chronically infected male's offspring and the role of the female for that litter? Assuming the female might become infected through the male, and then infection to the litter could be through the female?

S16: providing both the 'neutralization' and 'no-neutralization' is unnecessary and confusing. Also I would much prefer to see this data in a figure showing neutralization over time. Also S16 shows data from individuals from varying groups, lumped together; and not the total number of individuals from which antibodies were determined as indicated in table S8 shows many more blood values. How were the ones in Table S16 selected?

Version 1:

Reviewer comments:

Reviewer #2

(Remarks to the Author)

Reviewer #3

(Remarks to the Author)

Dear authors,

Thank you very much for your thorough answers to all of my concerns, and changes made if deemed necessary. All of my concerns have been addressed and I did not identify new ones. I think your study provides a significant contribution to our understanding of Lassavirus infection in Mastomys rats.

Reviewer #1 (Remarks to the Author):

Hoffmann, et al report on a much-needed systematic analysis of the persistence of Lassa virus (LASV) in one of the natural reservoirs for the virus; *Mastomys natalensis*. The authors provide excellent data on the numerous ways that LASV persistence can occur. Of particular note is the ability of *M. natalensis* to be persistently infected with LASV for over 16 months in a laboratory setting which goes well beyond the normal life span of these rodents. These data along with the horizontal and vertical transmission (male or female) shown in this manuscript brings home the importance of how much and for how long the transmission threat comes from *M. natalensis*.

Minor comments:

Line 79: What is the inoculation route? Could not find in methods either.

The Inoculation routes are described in Line 118 for the different age groups and Line 198 for prenatal infections, as well as in the Methods section in Line 582–589.

We added “subcutaneously (s.c.)” to Line 79 and abbreviated “subcutaneously” in Line 118 with “s.c.”

Antibodies surveyed within the study were directed at the nucleoprotein. Are there antibodies to the glycoprotein found and are they neutralizing at all?

The ELISA kit used in this study has been previously validated by our group for the use on rodent samples, showing it to be cross-reactive. However, the other commercially available ELISA kits in our repertoire that target GP-specific antibodies are all human specific and were not cross-reactive.

*We know in humans GP-specific antibodies, including neutralizing ones, can be found. Therefore, it is very likely that they are also present in *Mastomys*, however, we have not tested for them. We measured the neutralizing capacity of the antibodies and in line with human survivors, neutralizing antibodies only rise late after infection (described from line 414).*

Line 185: Why the change in inoculation route for the prenatal infection model?

Based on studies on LCMV, we only have a short time window to possibly achieve in-utero transmission by inoculating a pregnant female. Hence, we needed an inoculation route that would result in quick and systemic virus dissemination.

*Due to the pregnancy intraperitoneal was not an option, so we chose intravenously. Since adult *Mastomys* have very thick skin on their tails, we choose retro-orbital as the inoculation site.*

Reviewer #2 (Remarks to the Author):

Understanding how zoonotic arenaviruses such as LASV are transmitted and maintained in reservoir hosts is important to predict and prevent outbreaks. Only a few studies have looked at persistence of LASV in its natural host in laboratory experiments – especially young animals. This study examines both horizontal and vertical transmission of LASV in its natural host rodent species.

The work presented here is quite a thorough tour-de-force study of persistence in the natural reservoir species. In adults and animals older than 2-weeks, LASV infection resulted in transient infections with subsequent seroconversion. However, the

authors of the study here found that neonates and younger juveniles exhibited persistent infections lasting up to 16 months despite antibody presence. Intriguingly, despite detection of viral antigen in many tissues, no pathological lesions were found, and no clinical signs were observed in the animals. The persistently infected animals were able to transmit the virus throughout their lives, providing a new understanding of transmission dynamics in the natural host. The antibodies produced during infection appear to be mostly non-neutralizing, which provides an intriguing avenue for future exploration. The results here are similar to this research team's work with the related virus MORV whereby reservoir contact with MORV within the first two weeks of life led to virus persistence, whereas older animals only developed transient infections.

This is a very well-done study using a difficult experimental system (BSL-4 virus; wild rodent species) that provides important new information for the field. I particularly like the mouse organ heat maps in Fig 4-7.

My comments are minor:

1. How were LASV inoculation doses chosen?

*The inoculation dose of 1,000 FFU is commonly used by our group for LASV experiments with our chimeric mouse model. Since we already know the course of infection and disease of LASV for this dose in mice, it served as reference for the experiments with the natural rodent host. Furthermore, this inoculation dose has also been used other studies involving adult *Mastomys*, allowing for an easier comparison between the results from our experiments and those of other groups.*

For the inoculation of pregnant females, we have chosen 10,000 FFU i.v. to guarantee a quick systemic virus dissemination during the short time window between inoculation and birth.

2. Fig 5A – does each symbol represent a single animal tested at that time point, or were some animals retested longitudinally?

Most of the blood samples were collected at the time of euthanasia, however, some individuals were also sampled longitudinal. In the current version of the graph we have not visually distinguished the terminal and longitudinal samples in order to keep the number of different symbols used as low as possible. The Figures Fig 3 and Fig 5 have been amended and longitudinal samples are now indicated by a crossed-out symbol. The corresponding explanation “Blood samples acquired through terminal sampling are indicated by clear symbols, whereas longitudinal samples are indicated by crossed out icons” has been added to the figure descriptions (Line 161–162 and Line 217–218).

3. Fig 5 – can the authors confirm that transmission happened during pregnancy rather than after the pups were born?

For this experimental set-up we most likely observe a mix between in utero and post-natal infection. We could detect LASV in neonates as early as two days post birth, which based on the virus dynamics we observed during our experiments is rather unlikely to be caused by post-natal infection. However, in this early phase, not all tested individuals were LASV-positive and the number of infected animals in the litter increased over time, which points towards post-natal infection.

Based on the very brief periods of infection in adult animals it is unlikely that the mother is the source of infections that were acquired after birth and what we observe is most likely based on pup-to-pup transmission.

4. At what stage of gestation were the pregnant females euthanized (Line 221-222 and Fig 6)?

Pregnant females were euthanized within the first two weeks of gestation. The information “within the first two weeks of gestation” was added to Line 235.

5. Arrows on histo images in Fig 6C would be beneficial.

Arrows have been added to Fig 6C to highlight infected tissue sections. The sentence “Arrows serve as visual aid indicating infected cells.” Has been added to the Figure description in Line 266–267.

6. What counterstain is used in the IHC images?

The IHC images were counterstained with hematoxylin, as described in the Methods section Line 657–658.

7. Line 536-537 – virus stock used for these studies has an unknown passage history. Has it been sequenced?

Yes, the virus has been sequenced and the sequences are publicly available. When we acquired the virus stock, we got the information that it has been passaged less than 5 times, however, no exact number was given. At the BNITM the virus was passaged less than 3 times.

The Section in the manuscript has been amended and “has been passaged less than 10 times in total” has been added in Line 579. The following sections have been removed Line 577–578 “with unknown passage history”, and Line 580 “passaged less than 3 times at the BNITM.”

For further clarification the following section was added to the Appendix Line 24–25, “Virus strain. Experimental virus stocks are regularly checked for mycoplasma contamination by PCR and for the absence of mutations by Next Generation Sequencing.”

8. Fig 9 on the transmission model is confusing.

The core message of this figure is to visualize the age-dependent course of infection that we observed. The curved arrows were intended to indicate different routes of transmission. However, we agree that in the current form, this might be a bit unclear. We have removed the curved arrows and the corresponding text (“horizontal transmission”, and “Vertical/horizontal transmission”) have been removed in Fig 9.

Reviewer #3 (Remarks to the Author):

Key-findings

Hoffman et al., Lassa virus persistence with high viral titers following experimental infection in its natural reservoir host, *Mastomys natalensis* describes the results from a great number of experimental infections. The key result from these experiments is that virus infection that starts prior to the age of 2 weeks causes the mice to become persistently infected, while infection that starts after this age will be transient. Another key finding is that some infected animals remain so while they develop antibodies, including neutralizing antibodies. Lastly the virus infection could not be associated with disease in any of the infected animals.

Quality of the data:

The animal experiments seemed to have been performed with great care, and analyses thorough and detailed including serial and long-term clinical assessments, virology (qPCR and culturing), serology (ELISA and VNTs) and histopathology

including IHC. I really like LASV was investigated in its original host. This type of research can help to understand the virus infection in the reservoir much better and hence help in preventing disease but also understanding the disease in humans - comparative pathogenesis. It is often not easy to do these experiments in non-conventional animals in the laboratory. It is also not easy to follow up on the long time (16 month) so congratulations with this success. However, I do have some concerns about the set-up of the animal experiments.

- Virus strain: Was any sequencing performed to learn how the virus strain used for these experiments (that has an unknown passage history) was comparable to a wildtype virus – and thus can be used to model infection in the wild?

As mentioned in the response to Reviewer 2, who raised a similar question, in our hands the virus has been passaged less than 3 times. Prior to this we do not have an exact passage number, only that it has been less than 5 times. The manuscript has been amended and “has been passaged less than 10 times in total” has been added to Line 579. The sections “with unknown passage history” (Line 577–578) and “passaged less than 3 times at the BNITM” (Line 580) have been removed.

The LASV strain Ba366 used in this study has a sequence identity of >95.5 % (E value 0.0) with other strains collected from rodents in the same area in Guinea. All our virus strains are regularly checked for mycoplasma contaminations and mutations.

The section “Virus strain. Experimental virus stocks are regularly checked for mycoplasma contamination by PCR and for the absence of mutations by Next Generation Sequencing.” has been added to the appendix Line 24–25.

- Route of inoculation: this can have great effects on type and course of virus infections. Subcutaneous and intravenous routes were used, while authors themselves argue that this is not likely to be an important natural route of infection for LASV in Mastomys mice.

We are aware of the route specific differences that have been described in other studies. Nonetheless, we have chosen s.c. as uniform inoculation route because it allows efficient application of defined virus quantities for animals of various age and size. Furthermore, it is in line with previous experiments with non-pathogenic arenaviruses increasing comparability of data for different viruses. With this route it is also feasible to safely handle and inoculate small pups but also larger numbers of animals in a BSL4 context.

As described in the response to Reviewer 1, we have chosen the i.v. route to inoculate pregnant females, because it leads to an early systemic viral replication and would potentially allow trans-placental transmission to embryos within a short time window.

Our co-caging and breeding experiments were conducted to cover the “natural transmission” between infected individuals and their exposed contacts.

- Read-out: H&E and IHC were performed on persistently infected animals – at varying times post inoculation. If any conclusions are drawn by authors regarding a lack of tissue response to virus, authors should be very clear about which groups were actually investigated by these techniques. In my opinion these groups should be more carefully defined because there are several different variables that are likely to influence the pathogenesis or pathology, and therefore they should not be lumped into one 'persistently

infected' group. Relevant are at least the age at inoculation, duration of the infection, antibody presence at time of death.

This crucial information was indeed missing. We have now added additional information, including age, time since inoculation, as well as antibody status, on the animals that used for IHC and H&E stainings to the appendix. As described in the Results and Methods sections persistently infected animals that were chosen for the histological analysis had to have stable viral RNA titers in blood for ≥ 120 days. Furthermore, we have than selected animals covering the three experimental approaches that had the highest chance to result in persistent infections. Namely, animals inoculated at 6–7 days of age, categorized as “Inoculation”, and animals that were exposed to LASV in utero either through the inoculation of a female during gestation, categorized under “Prenatal Infection”, or those borne to persistently infected females, labeled as “Offspring”.

Detailed information on the animals for which IHC and H&E images are shown is now available in the new table S9 that has been added to the appendix Line 284. A reference to this table was added to the method section Line 661–662: “Detailed information on the animals for which IHC and H&E images are shown is available in Table S9.”

- A lot of data is presented- an impressive amount of work. However, I wonder if all are necessary, or necessarily in the right format to show the main outcomes of the study. I got lost between the different experiments and read-outs and data -see more detailed comments below. What was the aim and outcome of the transmission experiments? It did not become clear to me. Despite my concerns I think the main conclusion on the age of inoculation determining the likelihood of chronic LASV infections is correct based on the research findings described here. I welcome an explanation for the inoculation routes used – and why the infections in the experiment mimic the natural situation despite of it. The histology and immunohistochemistry photos clearly show the occurrence of widespread infection without clear development of lesions in those cases; however it is not clear what cases these examples represent. This could be made more clear.

In my opinion the main result, the effect of the age when Mastomys rats become infected, determining the further development of the infection, is significantly contributing to our understanding of the LASV reservoir.

The specific questions are answered in more detail in response to comments above and below. To summarize the reasoning for the transmission experiments: The major aim of this study was to describe the infection phenotype of LASV in its natural host. Emphasis was put on persistent LASV infections, its characteristics and whether LASV-persistence occurs in an age-dependent manner (similar to MORV-infections).

One of the overarching aims is to further the understanding of how LASV behaves in its natural reservoir in the wild. Since in the wild, LASV infections in Mastomys would be the result of exposure to infected individuals or their excretions, we conducted the complementary transmission experiments, which were intended to determine whether the infections we observed following inoculation are in fact transmittable to exposed individuals and what the infection phenotype in these exposed animals would be.

The breeding experiments were the continuation of the prenatal infection experiments. One important conclusion of the prenatal infection experiments is the fact that successful infection of the offspring following inoculation during

gestation is a very timing sensitive matter. The breeding with persistently infected males was the exposure-based pendant to the inoculation of pregnant females. Given the ubiquitous and stable virus presence in body fluids and organs of persistently infected animals, a persistently infected female would circumvent the timing issue. Furthermore, investigating mother-to-offspring transmission was the next logical step, considering the age-dependence of LASV infection.

Also, I have some more specific issues with clarity and data presentation, or questions /concerns, outlined below:

Please explain, if one aim of the experiments was to determine viral excretion sites and routes of transmission, why was feces tested, but not intestines (while the virus is obviously epitheliotropic). Could feces not be an excretion route, via intestinal epithelial cell infection?

This is a very good point, unfortunately intestines were not collected during our experiments and only a small number of feces samples was gathered, which does not allow a concrete conclusion on fecal excretions as transmission route.

The following section has been added to Line 431–433 “The observed epithelial tropism indicates the intestines, which have not been sampled during the experiments, as another potential target site for LASV infection and also suggests feces as a possible excretion route.”

Regarding the feces that was tested, there were only two (line 324); while 538 animals were used in the experiments? Not clear how difficulties of testing feces for infectious virus were overcome (as the sample usually negatively affects the cells on which virus is grown).

Unlike for urine, feces samples have not been collected on a regular basis. We have not collected terminal feces samples, rather droppings were collected during general handling if the sample could be definitively linked to specific individuals. As a result, the total number of feces samples is very small. As described in the Methods section, droppings were resuspended and homogenized in PBS and larger particles were removed through centrifugation. Furthermore, the cell culture medium used during the in vitro infectivity experiments contained penicillin (100 U/mL) and streptomycin (0.1 mg/mL). During the experiments no signs of bacterial or fungal growth were observed, and cells on the cover slips appeared viable.

The following sentence was added to Line 447–448 “Similarly, we were unable to isolate LASV from feces samples, despite the presence of RNA.”

The word “sporadically” has been added to the Method section Line 608 to highlight that feces samples have not been collected on a regular basis.

Line 18 and 19: 'However, neonates and younger juveniles' what is younger than a neonate? Consider to rephrase 'neonates and younger juveniles' to something like 'mice younger than 2 weeks' to be clear.

Line 18 and 19 have been amended, the section “neonates and younger juveniles” has been deleted and the sentence was rephrased to “mice younger than two weeks”.

Line 43: Please rephrase, as zoonotic transmission itself can not be driving an outbreak; people making contacts with animals can be seen as driving the occurrence of zoonotic transmissions, which can be the cause of the start of the outbreak.

Line 42–43 has been amended and “rodent-to-human contact” has been added.

Line74: 'organ pathology'

The word “organ” has been removed and the word pathology changed to “Pathology” in Line 74.

Line 90: size \diamond weight would be more precise here

The word “size” in Line 93 has been replaced with “weight” accordingly.

In figure 1 1-d: body weight on y axis instead of 'weight' would be more clear.

The title of the y-axis for graphs in Figure 1 and Figure S9 have been changed to “Body weight”.

In figure 1c I would challenge the statistics that there is no effect seen from the inoculation. Five dots after the challenge are below the black line indicating the trend, before weights recover again. Then there is also more variation in week 3.5-4.5 with dots being well above or below the black line in this group – more than in the other groups. Not certain what statistics should be used but I imagine one could calculate based on the other data how likely it is to have by just variation, those five dots in a row after the inoculation lower than expected based on the growth curves seen in all groups. Still, the effect is not necessarily from the infection, could also be from the inoculation as the negative control group were not sham inoculated.

This is a very good point, animals that were inoculated at day 15 were anesthetized for the procedure, whereas none of the other groups required anesthesia. The sudden body weight decline and fluctuation in the days immediately after inoculation could be a possible handling effect on body weight. Possible effects of the infection on body weight could be expected to manifest roughly one week post-infection, at which point the body weight already stabilized. The later deviation could be explained by sex differences that start to manifest from the age of four weeks onwards.

The following sentences have been added to Line 87–90 “Although statistically non-significant, we observed fluctuations in body weight of animals inoculated at 15 days of age during the first week following inoculation. However, animals recovered within a few days from the weight drop.” and Line 466–468

“Although, we did observe a short-lived drop in body weight following the inoculation of animals at 15 days of age which was most likely caused by handling induced stress during the inoculation procedure.”

Fig 2: Please provide a serial section with immunohistochemistry staining to show the presence of virus antigen in cells and tissues provided. As lesions are mostly expected to be associated with infected cells it is important for the interpretation to show them side by side.

We thank the reviewer for this comment, the focus of Fig 2 is purely to show the absence of lesions in animals that were inoculated, which was uniform across the different experimental groups, including animals that were already exposed to LASV in utero. Although they are not presented side-by-side the representative images of IHC and H&E stainings that are shown in the main text and the appendix are always matched, i.e. taken from the same individuals. Relevant sections in the Method part have been amended. Line 658 the word “Representative” was deleted, and the beginning of the sentence has been changed to “Images ...”. The following section has been added to Line 659–662 “Representative images of the IHC and H&E staining shown throughout this study are matched, i.e. taken from the same individuals, within the different experimental groups.”

In general for clarity, but eg Line 117: 'virus load was determined' it is important to make a difference between viral load measured as RNA by PCR, or

as infective viral particles by virus titrations, especially as you performed back calculations to copies/ml instead of reporting ct values it might not be clear what was actually measured and this is relevant for interpreting 'persistent infection' as being individual shedding infectious virus, or individuals being viral RNA positive for a very long time.

To make the distinction between the PCR- and titration-based titers clearer several phrases have been amended. Line 120 "virus load" was changed to "viral RNA load". Line 121 "organ titers" was changed to "virus titers in organs". Line 243 "virus titers" was changed to "RNA titers". Furthermore, the Label of the y axis has been changed from "Virus titer" to "Viral RNA" for the following figures Fig 3, Fig 5, Fig 7, Fig S7, Fig S17, and Fig S22. Furthermore, the phrase "Virus titers" has been changed to "Virus RNA titers" in the corresponding Figure descriptions (Line 156, 214, 275, as well as Appendix Line 102, 185, and 228).

Line 123, four individuals of how many in total?; also this sentence seems to contradict with the previous one as apparently not all animals remained viremic up to 34 weeks.

The word "most" was added to Line 126 and "out of 18" was added to Line 127.

Lines 132 /133: 'Some animals exhibited effective virus control and receding, or absent viremia was followed by virus clearance in organs.' Not sure what the results are that are described here. Some? How many of the total? The previous sentence suggests only persistent infected animals are described from that point onwards. How was decided they were persistently infected with absence of viremia measured?

The passage describes the two infection phenotypes that we observed following the inoculation at 6–7 or 11 days and how viremia and virus presence in organs were linked. In both experimental groups we observed animals that appeared to have developed persistent infections or went towards virus clearance. The main difference between the two groups was the percentage of animals that remained infected, which was lower for the day 11 inoculation. However, the general infection phenotypes were quite similar regarding viral RNA levels and virus titers in organs. The number of animals that showed stable viremia and thus stable virus titers in organs is described in Line 127–128 for the day 6–7 inoculation and in Line 130 for the day 11 inoculation.

The following changes were made to the text: In Line 130 "(1/3)" has been added. In Line 135 the phrase "In case of persistent viremia, organ titers remained stable over time and ranged" has been replaced with "Animals either inoculated at 6–7 or 11 days that displayed persistent and stable viremia also showed stable virus titers in organs over time, ranging ..."

The sentence "In contrast, the receding or absent viremia observed in animals that demonstrated effective virus control, was always accompanied by virus clearance in organs." was added to Line 138–139.

Also, from the methodology I understood that for tissue samples viral loads were determined by virus titrations and hence testing for infectious titers. Would be important to have results on how infectious titers and ct values compared in these organs of animals with antibodies to those without antibodies – to see if there was neutralization observed when culturing from the tissues of animals in which antibodies were detected.

We could not observe any differences in virus titers in organs of animals with neutralizing antibodies compared to animals without within the respective experimental groups. As we have described the neutralizing antibodies are only found in a few animals and appear quite late, they most likely only are present in a small subset of tested animals. Furthermore, given the high virus titers in organs, stained foci during the immunofocus assay were usually only countable in the 1:1000 or higher dilutions, whereas the highest plasma dilution with detectable neutralizing antibody response (IC₅₀) was 1:255. Organ samples were homogenized in 1 mL cell culture medium and serial dilutions were created for the titrations, which would also further dilute antibodies present in the samples to a most likely negligible level.

Line 144: 'LASV found in some organs for up to 2 weeks.' This is an example. Please throughout the results, including in figures, make clear what methodology was used to state virus presence, with IHC virus antigen, with PCR viral RNA or with culturing infectious virus? Or multiple techniques? This matters a lot for interpretation. For example RNA can be often found after viral infections while it is then not corresponding to aversive health effects or transmission.

As described in the response to a previous comment (“In general for clarity, but eg Line 117...”) we have amended the sections in the manuscript accordingly. Now all results pertaining qPCR-based analysis are referenced as either viremia, virus RNA load, or viral RNA titers etc., and any references to virus titers are now purely referring to immunofocus assay results. In Line 151 the word “LASV” has been replaced with “infectious virus”.

Fig 3: would prefer to see the a to e figures labeled with the group ID (inoculation time).

The Group ID has been added to the graphs in Figure 3. For consistency the Group IDs have also been added to Fig 1 a-d, Fig 3, Fig 4, Fig S9, and Fig S22.

Line 145: not clear what authors seem as 'limited' virus presence and what this is based on.

The rather ambiguous phrase “limited virus presence” in Line 153 was deleted and the sentence was changed to “infectious virus presence was only found in liver, spleen, lung, heart, and salivary glands”

Fig 4: Nice graphical way to show virus distribution in the different groups over time. However, what is the aim of the figure? I would think the main result shown is which cells or tissues contribute to the viral load present in the infected individual? Why was virus titer used as data for the figure, and not immunohistochemistry? IHC can be much more directly linked to virus in tissue. Immunohistochemistry is easier to interpret in relation to tissue, as with the other techniques you are also measuring virus in fluids. It is possible to quantify immunohistochemistry results in many ways. What are the arguments to use virus titer instead?

The immunofocus assay is a classical virological tool which gives a quantifiable readout for virus load which is commonly used for in vivo studies of LASV in our group and is frequently described in literature. It also allows the detection of infectious virus, which in theory could be isolated and used for further experiments.

Furthermore, the format of this assay allows a relatively high sample throughput making it possible to process large quantities of samples in a reasonable fashion. The analysis of the 2385 organ samples that were

collected during this study would not have been feasible using immunohistochemistry.

Could the results be influenced by the presence of antibodies in some of the animals?

We have not observed any differences in organ titers of animals with and without neutralizing antibodies within the respective experimental groups. Since we have not tested all animals for the presence of neutralizing antibodies, we cannot fully exclude a possible effect for all tested samples. However, as mentioned above only very few of the animals tested had neutralizing antibodies often at low titers and antibodies present in the organ samples would have been diluted during sample processing and the serial dilution for the immunofocus assay.

Visually: the difference between blue and black is not clear to me, why not just use black or blue as the interpretation is the same (no virus detected).

Negative samples have been assigned the default value of the limit of detection (LOD). As a result, organs where most samples tested negative and only few samples tested positive (possibly at low titers) are displayed in a blue color on the lower end of the scale close to the LOD.

The color black has been chosen to highlight the complete absence of infectious virus in any of the tested samples for a particular organ during a particular sampling period. This distinction is particularly important for the inoculation at the age of 28–29 and 57–59 days, in both instances we have only low levels of infectious virus in a very few organs, while the majority of organs tested completely negative.

Due to the very similar color tone of the dark blue of the LOD and the black, we have adjusted the scale for all heatmaps, i.e. shifting it more to purple, to increase the contrast. The following figures and figure descriptions were altered: Fig 4 (Line 173), Fig 5 (Line 224), Fig 6 (Line 267), Fig 7 (Line 285), Fig S7 (Appendix Line 111), and Fig S17 (Appendix Line 196).

For eyes it is not possible to see the color to see the viral titer. Again, I would prefer to have the group indication (so the inoculation time) indicated in the separate figures for clarity. How to deal with male/female and gonads in the figures as the n will be different for those organs.

To make the color in the eyes easier to identify, all schematic overview figures showing the organ titers, namely Fig 4, Fig 5, Fig S7, and Fig S17, have been rearranged and the head schematic has been enlarged.

The precise number of tested male and female gonads for each group during the sampling periods was broken down in the corresponding Supplementary tables and can now be found in the new supplementary figures (Fig. S2, S3, S4, S5, S6, S8, S11, S13, S14, and S18).

Line 172: 'LASV-infected individuals' please indicate at what time post inoculation the infected individuals are, and how the exposure was.

The neonates were exposed to their previously inoculated older siblings through co-housing from the moment they were born. At the time of birth, the older siblings were roughly 2–3 weeks post-inoculation.

For clarity the following changes were made: Line 180–181 was changed to “Three litters were exposed from birth via co-housing (n = 23) to infected older siblings” and the words “natural contact” were removed. In Line 182 “encountered their previously inoculated older siblings (2–3 wpi)” was added to the sentence and “encountered LASV-infected individuals” was removed.

The following changes were made to the Appendix: The info “(2–3 wpi)” was added to Fig S7 Line 102. The phrase “older siblings (2–3 wpi)” was added to the description of Fig S9 Line 124, the word “individuals” was removed. The footnote 3 “³ Neonates were exposed from birth via co-housing to infected older siblings (2–3 wpi).” has been added to Table S2 Line 253–254.

Line 179: 0.87 g per day, p = 0.0091; please indicate if this was 0.87 g per day growth, or 0.87 g per day extra growth compared to the non-exposed?

The number shown is the total per day growth of the infected animals. The Text has been amended accordingly and the body weight gain of control animals was added.

In Line 189, the word “faster” was removed, and the sentence (Line 188–191) was changed to “Exposed individuals did show a body weight increase of 0.87 g per day, which was higher (p = 0.0091) compared to the per day growth of the naïve controls (0.66 g per day)”. Furthermore, Line 257–259 has been changed to “Individuals borne to persistently infected females displayed an increased body weight gain of 0.8 g per day (p = 0.0066) compared to naïve controls at 0.66 g per day”.

Line 180: 'The parents of inoculated offspring showed only signs of a transient infection' what does this mean? Clinical signs? What? Or RNA detection in blood? What is meant with 'inoculated' here, I thought they were infected by exposure to infected animals?

The section refers to the parents of litters that were inoculated at 6–7, 11 or 15 days of age, which were co-caged and thus exposed to their inoculated offspring. To make this clearer the following changes were made:

In Line 178 the section “their litters which have previously been inoculated at 6–7, 11 or 15 days of age” was added to the sentence and the word “offspring” was removed. Furthermore, Line 191–193 was changed to “The parents that were exposed to their inoculated offspring developed antibodies but no viremia or virus presence in organs was detected.”, the words “of” and “showed only signs of a transient infection” in Line 192 were removed. Furthermore, the sentence “While LASV-specific antibodies could be detected in all exposed individuals, no virus was found in blood or organs of the tested animals” in Line 194–196 was removed due to redundancy.

Line 215: it is not clear where these 'persistently infected' females come from. Were these from the group that was inoculated at day 6 post partum?

As described in the Method section of the appendix in Line 40–42, the four initial persistently infected females originate from either a day 6 inoculation or were borne to a female that was inoculated during pregnancy.

The following information was added to the appendix: the total number of females used “(n = 4)” in Line 40 and how many were derived from which group “(n = 2)” and “(n = 2)” in Line 41–42.

Lines 291 and further: for interpretation it is important to describe the distribution of the virus positive cells in the tissues (individual, clusters, diffuse etc.). Also important to describe the distribution of the staining in different cell types (granular or diffuse, cytoplasmic etc.).

This was indeed missing and we now included this information in the text. The following passage was added to Line 321–328: “Interestingly, infected cells are not evenly distributed throughout infected organs. We could show that regardless of the target organ, mainly epithelial cells are virus-protein positive. Within one organ, individual positive cells were detected, however, the majority of the persistently infected cells appeared in clusters (e.g. see liver

Figure 8a). Virus protein is abundant in the cytoplasm but may also appear slightly granular. In virus-positive epithelial cells, the majority of signal is detected towards the plasma membrane (e.g. liver Figure 8a) and often in a highly polarized pattern (e.g. lung or epididymis Figure 8a and b)."

Line 336: 'Despite the presence of nAbs, 72.7 % of tested animals had still detectable

viremia (Table S17, S18 and S19).' For each percentage add number positive, and numbers tested (8/11).

We added the number of positives/tested to the percentage values. The following was added to Line 359 "(8/11)", Line 362 "(5/8)" and "(1/8)", as well as Line 365 "(2/17)".

Lines 400-402: 'Of note, the staining of the viral glycoprotein showed a highly polarized virus protein production in epithelial cells towards the lumen of the respective organs. Not sure to what results this is referring to? Which organs was this observed? Was is polarized within the cell? Or within the tissue?

Many of the described epithelial cells are just one cell layer lining the lumen...

We thank the reviewer for this comment and made it more clear in the text, now. Since epithelial cells seems to be the main target of persistent infection, this finding refers to most of the investigated organs, but was most prominent in the salivary glands, the lung (bronchial epithelium), the epididymis, and the uterine tube. In brief, as the reviewer pointed out, many epithelial cells are arranged in a (one-) layered fashion. Secretion of factors by these cells is usually directed toward the apical side of the cell, which is directed toward the lumen of the respective organ e.g. bronchial lumen, salivary ducts, or seminal ducts. In our project, faint cytoplasmic staining for LASV protein could be detected throughout a given epithelial cell. However, toward the apical side, very high amounts of virus protein could be detected that appeared to be close to the plasma membrane. Thus, we hypothesize that virus particles are produced and secreted by these cells in a polarized manner into the lumen of efferent ducts. This is in line with our findings of very high virus titers in different body fluids.

The following changes were added to the manuscript: Line 424 "Of note, the staining of the viral glycoprotein showed a highly polarized virus protein production in epithelial cells towards the lumen of the respective organs." has been removed. The passage "Of note, while epithelial cells often only showed a faint LASV glycoprotein positive staining in the cytoplasm, very high amount of virus protein could be detected towards the apical side of these cells. This is especially visible in the epithelial cells of the salivary glands, bronchial epithelium, epididymis, and uterine tube (see Fig. 8a,b and S7f). Thus, we hypothesize that virus particles are produced and secreted by these cells in a polarized manner into the lumen of efferent ducts in these organs." was added to Line 426–431. In Line 431 "The latter" has been replaced by "This".

Line 431: 'Antibodies with other non-neutralizing effector functions could, however, still contribute to virus clearance.' This confuses me. Was more viral clearance seen in individuals with non-neutralizing antibodies? Or what is this sentence referring to or explaining?

We could not observe any effect of the neutralizing antibody presence on LASV infections, regardless of whether animals developed persistent infections or achieved virus clearance. Since we observed the presence of neutralizing antibodies together with high levels of viremia and virus titers in organs, it appears that neutralizing antibodies on their own are insufficient to

aid in virus clearance. Nonetheless, some animals were able to clear their infection, which could in parts be aided by other antibody functions beyond neutralization. This is of course currently purely speculative, thus the corresponding sentence in line 465 “Antibodies with other non-neutralizing effector functions could, however, still contribute to virus clearance.” has been removed.

Line 581: 'Determination of virus titers and antibody status' There is lot's of comparisons between groups on virus titers per organ e.g.. How was the input controlled to make this comparison valid? E.g. weighing of the sample size or RNA output? There is no data provided on how the organ samples were pretreated (e.g. homogenizing etc) – although information of feces and swabs is present in 'sample pretreatment'.

Set volumes were used for the PCR set-up and the serial dilution of processed organ samples was used. Furthermore, for the calculation of viral RNA titers in body fluids, the original sample volume for RNA extraction, the volume of elution and the volume used for the PCR reactions were taken into account. Infectious virus titers (FFU/g) in organs were calculated using the weight of the original organ sample.

We have added the information on organ sample processing to the methods section. The sentence “Collected organ samples were homogenized in 1 mL cell culture medium using the FastPrep-24™ 5G tissue lysis system with the Lysing matrix D.” was added to Line 639–640. In Line 640 “Organ samples were processed for further analysis and” was deleted and the sentence was changed to “Infectious virus titers in organs were determined by immunofocus assay as described before⁴¹”.

Lines 467-469: 'LASV has also been detected in stomachs of suckling pups implying the possibility for transmission via breast milk.' I don't see a reference for that? Please be aware that pups in utero will drink amniotic fluid, and that could also be an explanation for finding virus in the stomach.

This is a valid point, that has been overlooked; we have amended the text in Line 507 with the following “post-birth or via swallowing amniotic fluid in utero”.

Starting line 480: 'suggesting that the exposure to LASV from early embryonic development leads to the acquisition of a central immune tolerance that also encompasses LASV-reactive B cells.' Please explain the term 'central immune tolerance' or provide a reference. And later also 'immune tolerance mechanisms'. What mechanisms? It is not clear how 'Once passaged across several generations the humoral response became completely absent' suggests 'the exposure to LASV from early embryonic development leads to the acquisition of a central immune tolerance that also encompasses LASV-reactive B cells.' When virus replicates in a fetus in utero prior to positive and negative selection of T and B cells, the T and B cell repertoire might not recognize the virus proteins as being non-host, and therefore no antibodies nor T cell immune response will occur. Is that what you are referring to? Please explain more, and also provide an explanation for how this increases with each passage.

The most likely explanation for the observed age-dependent development of persistent infections would be the acquisition of either a central or peripheral immune tolerance towards LASV antigens due to the exposure at a young age. For the animals that were borne to persistently infected females and thus have already been exposed to LASV in utero, a central immune tolerance, that encompasses T and B lymphocytes, would explain the absence of virus

clearance and the humoral response. Further explanations and references for the immune tolerance have been added to the text.

The humoral response we detected is likely caused by the presence of maternal antibodies that fade away over time. While there are still some individuals that show a lasting humoral response, most individuals in the first generation appear to be unable to produce antibodies on their own and thus could not pass down maternal antibodies to their offspring, further reducing the percentage of sero-positive animals in the next generation and subsequently leading to the complete absence of a humoral response.

The following changes were made to the manuscript:

Line 399 “and the development of a central or peripheral immune tolerance due to the exposure to LASV at a young age³⁸” has been added. Line 401 “tolerance” has been replaced with “persistence”. A new Reference has been added to the text (Line 400) and the reference section in Line 795–796 “Salaman, M. R. & Gould, K. G. Breakdown of T-cell ignorance: The tolerance failure responsible for mainstream autoimmune diseases? *J Transl Autoimmun* 3, 100070 (2020).”

The section in Line 518–522 has been changed to “The lack of a humoral response and the absence of effective virus control in these animals further suggest that the exposure to LASV from early embryonic development leads to the acquisition of a central immune tolerance, i.e. the depletion of virus-reactive T and B lymphocytes during early development^{38,52}”. In Line 522 “that also encompasses LASV-reactive B cells” has been removed.

A new reference (52) has been added to Line 522 and to the reference section Line 825–826 “Nemazee, D. Mechanisms of central tolerance for B cells. *Nature Reviews Immunology* vol. 17 281–294 Preprint at <https://doi.org/10.1038/nri.2017.19> (2017).” In Line 524 “mechanisms” has been removed.

Paragraph starting line 484: I don't understand the comparison. Is the question not rather if males can infect adult females via semen, and via adult females that are pregnant, fetuses can become infected? I don't understand how transmission from semen to offspring is relevant if the role of the transmission to the mother is not considered.

These are valid points raised by the reviewer. The reference to the persistence in the male reproductive tract was indeed intended to highlight the male reproductive tract and its excretions as a possible transmission route to females. The following changes were made for clarification: In Line 526 the beginning of the sentence was changed from “Since we ...” to “We...” and “suggesting excretions of the male reproductive tract as a possible transmission route to naïve females.” was added to Line 528–529.

The aim of the breeding experiments with persistently infected males was to assess whether we could achieve transmission from an infected male through the female to the unborn offspring, mirroring the inoculation of pregnant females. In Line 529–530 was changed to “We further assessed transmission from persistently infected males to their partners and via the female to their unborn progeny in the absence of direct contact.”

Authors do not discuss or explain how some animals with detectable neutralizing antibodies can remain infectious over a long time. Does the virus escape immunity by hiding at immune privileged sites and re-infecting other sites from there? Does the virus escape immunity by mutations?

This is a very good question, which we cannot fully answer currently. In contrast to Morogoro virus, persistent LASV infections are characterized by a continuous and ubiquitous virus presence in all organs and does not show fluctuations in titers over time, thus, a retreat to immune privileged sites and subsequent re-infection seems to be unlikely.

Currently we do not have information on possible mutations that occur within the host, however, the intra- and inter-host evolution of LASV over long-lasting infections and serial passage through infected animals is one of the potential future topics we would like to investigate. Presumably the antibody titers are too low to lead to virus clearance in the absence of other adaptive immune functions.

The following sentence was added to manuscript Line 464–465 “It is possible that nAbs appear too late and at too low quantities to cause virus clearance on their own without the aid of other immune cells”.

Supplementary data

Line 31: 'Two whole litters per experiment group were inoculated to account for the genetic variations of the parents.' I don't think that two breeding pairs can be seen as accounting for genetic variations. Rather I would state that animal experiments for ethical reasons should be balanced and that you have designed the experiments with the assumption that these two breeding pairs could be seen as representative.

Thanks to the reviewer for raising this point. We have amended the manuscript according to the reviewers suggestions.

In appendix Line 33–34 the words “to account for the genetic variations of the parents.” was deleted and the sentence changed to “Two whole litters per experiment group were inoculated.” Furthermore, the passage “In order to reduce the number of animals needed the inoculation experiments were designed based on the assumption that two breeding pairs per experimental group are representative of the different genetic backgrounds within our outbred colony.” has been added to Line 34–37.

lines 35-36: 'Persistently infected females used in the breeding experiments originated from either day 6 inoculation or the inoculation of a pregnant female' I have missed the description of infection dynamics after inoculation during pregnancy – how long were these followed for virus presence? Does inoculation during pregnancy consistently result in a chronic >40 week infection? Describing the infection dynamics after inoculating pregnant females is quite an important. These females were >12 weeks old, and it doesn't fit the pattern described in the other parts that they become persistently infected.

As described in a previous response, adult females that were inoculated during gestation only showed transient infections. The resulting offspring, however, was able to develop persistent infections. Thus, the persistently infected females used for breeding were borne to a female inoculated during gestation. The current phrasing was obviously not clear enough and misleading. For clarity Appendix Line 41 has been amended, “the inoculation of a pregnant female” was removed and replaced with “were borne to a female inoculated during pregnancy”.

Table S2: Why only min-max, and not an average or mean provided? Rather than a table I would like to see these kind of data in bar charts, similarly organized (per organ, and per group – together) so that the viral loads over time for each organ becomes clearer.

Since we already display mean-titers in the schematic heatmap overviews, we have chosen min-max values for the supplementary tables to show the range in which virus titers moved in a concise manner.

As per the Reviewers request, we have now replaced the supplementary tables with bar graphs showing virus titers in organs over time. These graphs display all data points for the individual organ samples and also show the mean titer. All references to the original tables have been changed accordingly. Furthermore, the numbering of all supplementary figures and all corresponding references in the text have been altered. The following changes were made:

Appendix

Deleted supplementary tables: Table S2 (Line 172–173), Table S3 (Line 174–175), S4 (Line 176–178), S6 (Line 192–193), S7 (Line 194–195), S9 (Line 203–204), S10 (Line 205–206), S11 (Line 207–208), S12 (Line 209–210), S13 (Line 211–213), S14 (Line 214–215), S15 (Line 216–218).

Inserted new supplementary figures and figure descriptions: Fig. S2 (Line 75–79), Fig. S3 (Line 80–84), Fig. S4 (Line 85–89), Fig. S5 (90–94), Fig. S6 (Line 95–99), Fig. S8 (Line 116–121), Fig. S11 (138–143), Fig. S12 (Line 144–149), Fig. S13 (Line 150–156), Fig. S14 (Line 157–163), Fig. S15 (Line 164–173), Fig. S18 (Line 200–206).

Changed numbers of remaining supplementary figures: Fig. S7 (Line 101, former S2), Fig. S9 (Line 123, former S3), Fig. S10 (Line 132, former S4), Fig. S16 (Line 175, former S5), Fig. S17 (Line 182, former S6), Fig. S19 (Line 208, former S7), Fig. S20 (Line 219, former S8), Fig. S21 (Line 223, former S9), Fig. S22 (Line 228, former S10).

Changed numbers for remaining supplementary tables: Table S2 (Line 246, former S5), Table S3 (Line 259, former S8), Table S4 (Line 267, former S16), Table S5 (Line 270, former S17), Table S6 (Line 274, former S18), Table S7 (Line 278, former S19), Table S8 (Line 281, former S20).

Main text

Changed references to supplementary figures and tables: Line 135 now (Fig. S2; Fig. S3), Line 147 now (Fig. S4), Line 151 now (Fig. S5), Line 154 now (Fig. S6), Line 184 now (Fig. S7a; Table S2), Line 188 now (Fig. S7b; Fig. S8), Line 191 now (Fig. S9a), Line 195 now (Table S2), Line 200 now (Fig. S9b), Line 201 now (Fig. S10), Line 204 now (Table S3), Line 205 now (Fig. S11), Line 237 now (Fig. S12), Line 244 now (Table S3), Line 251 now (Fig. S13 to S15), Line 255 now (Fig. S16), Line 257 now (Fig. S9c), Line 295 now (Fig. S17a), Line 296 now (Fig. S18; Table S13), Line 300 now (Fig. S17b), Line 304 now (Fig. S9d), Line 313 now (Fig. S19a), Line 315 now (Fig. S19a-d), Line 318 now (Fig. S19e,f), Line 319 now (Fig. S19a), Line 327 now (Fig. S20; Fig. S21), Line 339 now (Fig. S22a), Line 340 now (Fig. S22b), Line 346 now (Fig. S22c,d), Line 355 now (Table S4), Line 358 now (Table S5, S6, and S7), Line 573 now Table S8.

Table S5: It is not 'natural', it is an experiment. 'Exposure to infected parent since birth' or something similar is more precise. 'Exposure of adults to infected individuals ' Adults at what age(s)? What is not clear for this experiment is: At what stage(s) of their infection were the individuals used to expose the adults? What were their excretion routes, of those selected infected individuals? What was their immune status at the moment of the exposure experiment? How long were adults exposed to the infected individuals and via

what routes? How come the data from 16 to 52(!) weeks is now lumped together?

For clarity the title of Table S2 (former S5) in Appendix Line 246 has been amended with the following “Virus RNA presence in body fluids, antibody status, and infectious virus in organs” and the section “Overview of the infection status” has been deleted.

For consistency, these changes (deletion of “Overview of infection status” and addition of “Virus RNA presence in body fluids, antibody status, and infectious virus in organs”) have also been added to the following tables: Table S1 (Line 238–239), Table S3 (Line 259–260).

The title of the Group “Natural exposure since birth” in Table S2 has been changed to “Exposure to infected individuals since birth”.

Footnote 4 “⁴ Naïve adults were exposed to infected individuals from the moment of inoculation of their offspring (at 6–7, 11, or 15 days of age) and remained in direct contact for 1–3 weeks until the weaning of the young at the age of 3–4 weeks. Females that gave birth to another litter remained in contact for 4–6 weeks. Naïve adults that were paired with persistently infected partners remained in direct contact with their mate until the end of the breeding experiments (4–21 weeks).” has been added to Table S2 Line 254–258 to specify the time of exposure for adults.

All parental adults were negative for LASV prior to being exposed to their inoculated offspring. For our inoculation experiments, we used breeding pairs with adults of varying ages that were acquired from the breeding facility and moved to the BSL4 laboratory prior to the experiments. On occasion pairings were set up specifically for an experiment, these consisted mainly of younger adults (which had to be older than 12 weeks). However, due to easier handling and less loss of offspring due to stress-induced parental aggression, we mostly used more experienced breeding pairs that already produced several litters before they were included in experiments and were “retired” from the official breeding in the animal facility. These pairs usually consisted of older individuals, resulting in the large age range. The results for the adult group have been pooled, since we saw no variation between the individuals (all were negative) based on their age at the time of sampling.

For table S8: please provide information in the legend on what organs were tested.

The list of organs that were sampled has been added to the footnotes of the supplementary tables. Table S1 Line 243–244 “The following organs were sampled: liver, spleen, kidney, heart, lung, gonads, salivary glands and thymus.” Table S2 Line 251–252 “The following organs were sampled: liver, spleen, kidney, heart, lung, brain, gonads, and salivary glands.” Table S3 (former S8) Line 263–264 “The following organs were sampled: liver, spleen, kidney, heart, lung, brain, gonads, salivary glands, eyes, thymus, stomach, prostate, seminal glands, cervical and inguinal lymph nodes, placenta, blastocysts, whole embryos, and fetal livers.”

Table S10: chronic male and chronic female are used, as well as persistently infected parent. Please be precise and consistent with terminology. It is not clear what the table is referring to. How can one differentiate between chronically infected male's offspring and the role of the female for that litter? Assuming the female might become infected through the male, and then infection to the litter could be through the female?

Thanks to the reviewer for pointing out the inconsistency. As mentioned in a previous comment, Table S10 has been deleted. Nonetheless, the points raised still apply to the corresponding summary Table S3 (former S8). To keep the phrasing consistent, the word “chronic” has been replaced with “persistently infected” for all groups in Table S3.

As mentioned earlier in the response to the comment on “Paragraph starting line 484...”, the breeding experiments with persistently infected males were set up under the premise that they would infect their mate and through the female, the virus could be passed on to the offspring in the absence of direct contact between the male and the newborn. Footnote 3 was added to Table S3 Line 265–266; “³ Persistently infected males never had any direct contact to their offspring. The males served as the original source of the virus, and through infecting their female mate, the virus was passed down to the offspring.”

S16: providing both the 'neutralization' and 'no-neutralization' is unnecessary and confusing. Also I would much prefer to see this data in a figure showing neutralization over time. Also S16 shows data from individuals from varying groups, lumped together; and not the total number of individuals from which antibodies were determined as indicated in table S8 shows many more blood values. How were the ones in Table S16 selected?

The reviewer is right regarding the redundancies described for Table S4 (former S16). To avoid unnecessary confusion, we have removed the lines for “non-neutralization” in the table. Furthermore, “potentially exposed in utero, i.e.” was added to the footnotes in Line 268.

We performed the neutralization assays to gain a general overview on the presence of nAbs across the different experimental groups. Given the few instances of nAb detection and that their presence does not appear to aid in virus clearance, we did not perform any further neutralization assays for any other samples beyond this initial test.

Individuals were categorized into three general groups covering our different experimental approaches (inoculated animals, potential exposure in utero, and animals exposed to infected individuals), and samples were selected to cover rough time periods starting from the first appearance of antibodies.

We have not displayed the results in a figure showing neutralization over time, since the samples are not longitudinal.